# RefineEvo: Planning-Guided Heuristic Evolution with Bidirectional Experience

**Yang Wu** [* 1 2]  **Junran Pan** [* 1 2]  **Yifan Zhang** [1 2 3]  **Ning Xu** [1]  **Fanshuo Zeng** [1 2]  **Jian Cheng** [1 2 4]

## Abstract

Automatic Heuristic Design (AHD) has emerged as a transformative approach for solving combinatorial optimization problems. While recent Large Language Model (LLM)-based methods have shown promise, they predominantly rely on fixed evolutionary operators and struggle to effectively accumulate and reuse historical search experience. This paper proposes RefineEvo, a novel evolutionary framework that transforms AHD from a static trial-and-error process into a planning-guided, experience-driven system. RefineEvo introduces a Planner to dynamically schedule evolutionary operators and trigger refinement based on the current search state, and a Reflector to distill valuable lessons into a Bidirectional Experience Pool containing both positive insights and negative pitfalls. This synergistic framework enables the system to adapt its search tools to the evolving complexity of the problem and leverage trajectory-aware, situation-conditioned insights to guide generation. Experiments on several classic combinatorial optimization benchmarks demonstrate that RefineEvo consistently outperforms strong baselines. In particular, RefineEvo delivers superior solution quality while improving token efficiency, enabling more efficient and autonomous heuristic design.

## 1. Introduction

Automatic Heuristic Design (AHD) has emerged as a promising avenue for automating the discovery of effective heuristics, alleviating the burden of labor-intensive manual design (Burke et al., 2013; Stützle & López-Ibáñez, 2018). The recent integration of Large Language Models (LLMs)

[1]C²DL, Institute of Automation, Chinese Academy of Sciences [2]School of Artificial Intelligence, University of Chinese Academy of Sciences, Beijing [3]University of Chinese Academy of Sciences, Nanjing [4]AIRIA. Correspondence to: Yifan Zhang <yfzhang@nlpr.ia.ac.cn>.

*Proceedings of the 43ʳᵈ International Conference on Machine Learning*, Seoul, South Korea. PMLR 306, 2026. Copyright 2026 by the author(s).

further accelerates this process (Romera-Paredes et al., 2024; Liu et al., 2024b) by enabling semantic, structure-aware code transformations (Chen et al., 2021; Li et al., 2022; Achiam et al., 2023) that can generate genuinely novel algorithms. However, this capability also opens up an effectively unbounded program space (Wu et al., 2024; Liu et al., 2024c; Dong et al., 2025), making it crucial to harness LLMs to find superior solutions with minimal trial-and-error and computation.

Evolutionary operators in LLM-based AHD are typically implemented as prompt templates, ranging from exploration-oriented prompts that guide the LLM to mutate or crossover code to refinement-oriented prompts that improve existing solutions. Pioneering works (Romera-Paredes et al., 2024; Liu et al., 2024b) have compellingly demonstrated the effectiveness of coupling LLMs with evolutionary search by maintaining a fixed pool of pre-defined operators and applying them broadly across candidate algorithms. However, this "use-all" strategy overlooks operator heterogeneity and state-dependent utility (Di Tollo et al., 2015; Pei et al., 2025), so many operator applications are misaligned with the candidate's current stability and progress. Furthermore, as the evolutionary process advances, finding valid improvements becomes increasingly challenging (Ye et al., 2025; Cui et al., 2025). This necessitates the continuous refinement of the operators themselves to adapt to the evolving complexity of the search space.

Parallel to the focus on operators, complementary efforts have been directed towards accumulating experience from past iterations (Ye et al., 2024; Dat et al., 2025; Chen et al., 2025; Liu et al., 2025). These methods primarily employ outcome-based summarization, where insights are derived simply by contrasting superior and inferior candidates. However, this comparison treats algorithms as isolated items and ignores the fact that each solution is derived from a specific parent. Consequently, the system fails to capture the relative improvement of the offspring compared to its parent. Moreover, by purely generalizing from success, this approach ignores the situation-based nature of experience. That is, applying extracted experience is beneficial only when a particular situation is present. Without this context, the system treats limited experience as universally valid, leading to misleading guidance when applied to incompatible states (Xiong et al., 2025; Tan et al., 2025).

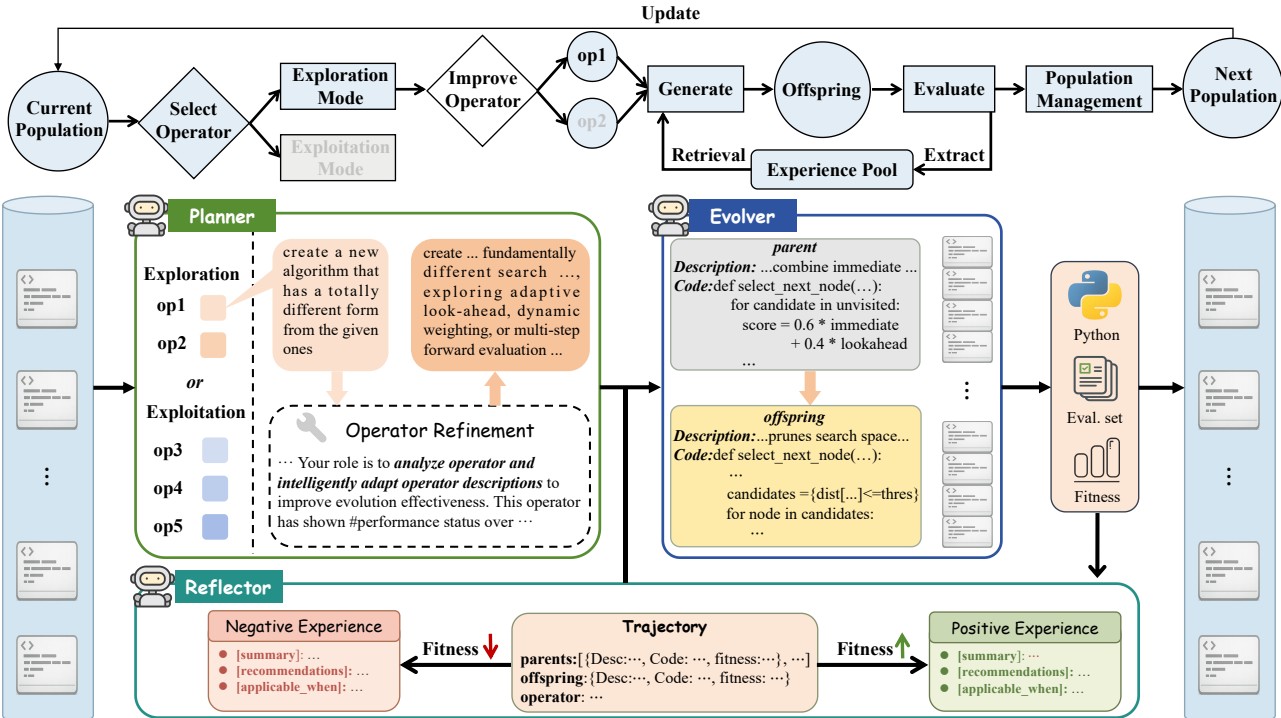

*Figure 1.* The overall framework of RefineEvo. At each generation, the Planner perceives population and operator statistics to select a search mode and trigger operator refinement when needed. The Evolver generates offspring using the selected operators and retrieves relevant experiences from the Bidirectional Experience Pool (BEP). The Reflector evaluates parent-to-offspring trajectories and distills structured experiences into the BEP.

To bridge these gaps, we present RefineEvo, a multi-agent framework to transform AHD into a self-evolving system. Specifically, we introduce a Planner agent to replace random trials with state-aware decision-making, identifying the most suitable operators based on the evolutionary status. When the search stagnates, the Planner triggers operator refinement to iteratively adapt the search tools, enabling them to transcend existing limitations and address increasing complexity. Finally, a Reflector distills parent-to-offspring trajectories into a Bidirectional Experience Pool (BEP), capturing both successful strategies and failure modes while binding them to their applicable preconditions. We demonstrate that this synergistic approach enables the discovery of superior heuristics, achieving state-of-the-art performance across diverse combinatorial optimization tasks.

In summary, our contributions are as follows:

1. We propose a Planner that performs state-aware operator selection and triggers dynamic operator refinement when progress stalls. This enables efficient navigation of the search space while adapting search tools to evolving complexity.

2. We introduce a Bidirectional Experience Pool that captures both positive and negative insights with trajectory-

aware, situation-conditioned signals. By binding experiences to their applicable preconditions, we resolve the ambiguity of outcome-based summarization and provide precise guidance for future generations.

3. We conduct extensive experiments on several classic combinatorial optimization problems across multiple algorithmic paradigms. RefineEvo consistently outperforms state-of-the-art baselines while achieving superior token efficiency.

## 2. Related Work

### 2.1. LLM-based Automatic Heuristic Design

The use of Large Language Models for heuristic design is a rapidly growing field (Romera-Paredes et al., 2024; Liu et al., 2024b; Huang et al., 2025; Ha et al., 2025; Yang et al., 2025; Novikov et al., 2025; Ghimire et al., 2025). Despite progress, many frameworks still rely on evolutionary search with largely static prompt-based operators (Liu et al., 2024b; Ye et al., 2024; Dat et al., 2025). EoH applies predefined strategies uniformly across generations (Liu et al., 2024b), and ReEvo follows a fixed evolution pipeline with limited explicit operator scheduling (Ye et al., 2024). MCTS-AHD (Zheng et al., 2025) structures exploration via tree search, yet it selects which heuristic node to ex-

pand rather than which operator to invoke, relying on a fixed action set for node expansion. LLM-LNS (Ye et al., 2025) evolves prompt strategies when stagnation is detected, but operates at a coarse, strategy-population level rather than diagnosing individual operator failures. In contrast, RefineEvo introduces a Planner that performs state-aware scheduling and per-operator conditional refinement, guided by operator-specific failure patterns. A detailed functional comparison is provided in Appendix A (Table 10).

## 2.2. Experience Mechanisms

Recent research demonstrates that augmenting LLMs with experience enhances reasoning (Shinn et al., 2023; Zhang et al., 2025a; Cai et al., 2025), while highlighting the critical role of relevance and quality control (Packer et al., 2023; Xiong et al., 2025; Tan et al., 2025; Ke et al., 2025). Within AHD, frameworks like ReEvo and HSEvo accumulate insights to guide evolutionary search (Ye et al., 2024; Dat et al., 2025). Concurrent preprints explore related directions such as foresight and hindsight prompting (Chen et al., 2025) and prompt-heuristic co-evolution (Liu et al., 2025). However, these approaches rely on outcome-centric summarization without explicitly storing the parent-to-offspring trajectory for retrieval, resulting in insights that are weakly grounded to specific applicability conditions. RefineEvo addresses this by extracting trajectory-aware experiences and actively managing the pool to ensure contextually aligned retrieval.

## 3. Methodology

Figure 1 illustrates the overall workflow of RefineEvo, a framework that transforms AHD from static trial-and-error into a planning-guided, experience-driven process. Operating as a closed loop, a Planner dynamically schedules and refines evolutionary operators based on the search state, while a Reflector distills historical trajectories into a Bidirectional Experience Pool (BEP) to guide the Evolver in generating experience-augmented offspring. In the following, we begin by formalizing the problem setup in Section 3.1, then detail the planning-guided evolution mechanism in Section 3.2, and finally describe the bidirectional experience mechanism in Section 3.3.

### 3.1. Preliminaries

Finding a heuristic $h$ that optimizes performance over a target problem distribution $\mathcal{I}$ can be formulated as maximizing the expected fitness

$$h^* = \arg\max_{h \in \mathcal{H}} F(h) = \arg\max_{h \in \mathcal{H}} \mathbb{E}_{I \sim \mathcal{I}}[f(h, I)], \quad (1)$$

where $\mathcal{H}$ denotes the program space. The fitness $F(h)$ is a scalar score quantifying the heuristic quality. For minimization tasks, it is defined as the negative cost.

Each heuristic is represented as a tuple $h = (d, c)$, where $d$ is the natural-language description and $c$ is the executable code (see Figure 1).

The search process maintains a population $\mathcal{P}_t = \{h_1, \ldots, h_N\}$ of size $N$ at each generation $t$. The population evolves via an elitist greedy protocol. After generating offspring $\mathcal{P}'_t$, the next population $\mathcal{P}_{t+1}$ retains the top-$N$ individuals from $\mathcal{P}_t \cup \mathcal{P}'_t$ based on fitness values.

### 3.2. Planning-Guided Evolution

Instead of applying a fixed set of evolutionary operators indiscriminately, RefineEvo employs a Planner to dynamically manage the search process by perceiving the evolutionary state, scheduling appropriate operators, and refining them when they become ineffective.

**Perception and Decision.** At generation $t$, the Planner perceives the global search state $\mathcal{S}_t = (\Phi_{\text{pop}}, \Psi_{\text{ops}})$. Here, $\Phi_{\text{pop}}$ captures population-level statistics, specifically fitness convergence (the stagnation trend of top performers) and population diversity. Simultaneously, $\Psi_{\text{ops}}$ tracks the recent utility of each operator, summarizing metrics such as the validity rate of generated code and the success rate in producing offspring that outperform their parents. Synthesizing these signals, it determines the optimal search mode rather than relying on a single fixed numeric threshold. When $\Phi_{\text{pop}}$ indicates premature convergence or diversity collapse, the system enters Exploration Mode, prioritizing operators that introduce structural novelty. Conversely, when promising seeds are identified (e.g., the best fitness improves steadily while top heuristics remain valid), it switches to Exploitation Mode, focusing on local refinement.

**Adaptive Operator Selection.** The Planner executes the chosen strategy by selecting a subset of operators from a predefined library $\mathcal{O}$. This library contains five meta-operations spanning exploration (e.g., generating distinct algorithms) and exploitation (e.g., parameter tuning); detailed definitions are provided in Appendix E.2. Within the active mode, it further prioritizes operators based on their historical utility recorded in $\Psi_{\text{ops}}$. This two-level selection mechanism aims to allocate the computational budget to tools that are both logically consistent with the current search phase and empirically proven to be effective.

**Dynamic Operator Refinement.** The Planner continuously monitors operator performance to address a critical limitation in static AHD methods: fixed prompt templates often lose effectiveness as problem complexity evolves. If an operator consistently fails to improve over the parent heuristics or yields a high invalidity rate for consecutive uses, it triggers a Refinement Event. During this process, it rewrites the natural language prompt—transforming generic

instructions into context-specific guidelines—based on the failure patterns observed in $\mathcal{S}_t$, along with the operator's recent performance statistics $\mathbf{z}_t$ (e.g., success rate and validity rate). Table 13 illustrates how generic initial descriptions evolve into specific, context-aware instructions through this mechanism.

### 3.3. Bidirectional Experience Pool

To address the limitation in existing AHD methods where only final elites are retained while valuable lessons from failed attempts are discarded, RefineEvo introduces a Bidirectional Experience Pool (BEP). This module explicitly maintains both positive insights (successes) and negative pitfalls (failures), coordinating a Reflector to accumulate experience and an Evolver to retrieve it.

**Instance-Level Situational Reflection.** The Reflector operates as a post-evaluation analyst to extract actionable insights. Instead of treating algorithms as isolated data points, it analyzes the evolutionary trajectory by comparing the offspring with its parent. If the offspring achieves a significant performance gain, the Reflector extracts the key modification strategy and records it as a positive experience to be reinforced. Conversely, if the offspring suffers a notable performance degradation, it identifies the erroneous change and records it as a negative experience to be avoided. Crucially, the Reflector binds these insights to specific contexts, preventing the misuse of generalized advice in incompatible situations.

**Experience Representation.** To ensure the gathered knowledge is actionable, each experience record is structured into three logically distinct components: a summary of the algorithmic modification, a recommendation stating what to do or avoid, and an applicable condition describing when this insight is relevant. This structured representation allows the system to distinguish between universally valid rules and situational tactics. Concrete examples of these records are provided in Appendix E.4.

**Contextual Retrieval-Augmented Evolution.** Before generating new heuristics, the Evolver utilizes the BEP to inform its decision-making through a context-aware mechanism. The Evolver first formulates a query based on the current search state, the selected operator's goal, and the characteristics of the parent heuristic. It then retrieves the most relevant records from the pool based on semantic similarity. Positive experiences are retrieved to provide "Insights to Follow," while negative experiences are retrieved to provide "Pitfalls to Avoid." These retrieved insights are explicitly injected into the generation prompt. This dual retrieval mechanism aims to help the Evolver reinforce effective patterns while avoiding repeated failure modes.

To prevent noisy or stale lessons from dominating retrieval, we maintain a lightweight maintenance policy that tracks each experience's utility based on downstream outcomes. Experiences with low utility are periodically pruned to keep the pool concise and high-quality (details in Appendix E.4).

We summarize the complete process of RefineEvo in Algorithm 1, illustrating how the Planner, Evolver, and Reflector coordinate across generations.

---

**Algorithm 1** RefineEvo

1: **Input:** instance set $\mathcal{I}$, generations $T_{\max}$, population size $N$, operator library $\mathcal{O}$, prompts $\{p_o\}_{o \in \mathcal{O}}$
2: **Output:** Best heuristic $h^*$
3: Initialize experience pools $\mathcal{E}^+, \mathcal{E}^- \leftarrow \emptyset$
4: /* Initialize (Sec. 3.1)            */
5: $\mathcal{P}_0 \leftarrow \text{INITPOPULATION}(N)$
6: Evaluate $F(h)$ for all $h \in \mathcal{P}_0$ on $\mathcal{I}$
7: **for** $t = 0$ **to** $T_{\max} - 1$ **do**
8:    /* Operator Selection (Sec. 3.2)     */
9:    $\mathcal{S}_t \leftarrow \text{GETSTATE}(\mathcal{P}_t, \mathcal{O}); \mathcal{P}'_t \leftarrow \emptyset$
10:   $mode_t, \mathcal{O}_t \leftarrow \text{PLANNER}(\mathcal{S}_t)$
11:   **for each** $o \in \mathcal{O}_t$ **do**
12:      /* Operator Refinement (Sec. 3.2) */
13:      **if** $\text{PLANNERTRIGGERREFINE}(o, \mathcal{S}_t, \mathbf{z}_t)$ **then**
14:         $p_o \leftarrow \text{REFINE}(p_o; \mathcal{S}_t, \mathbf{z}_t)$
15:      **end if**
16:      **for** $j = 1$ **to** $N$ **do**
17:         /* Retrieve Experience (Sec. 3.3) */
18:         $Parents \leftarrow \text{SELECTPARENTS}(\mathcal{P}_t, o)$
19:         $q \leftarrow \text{BUILDQUERY}(Parents, o, \mathcal{S}_t)$
20:         $C_{\exp} \leftarrow \text{RETRIEVE}(\mathcal{E}^+, q) \cup \text{RETRIEVE}(\mathcal{E}^-, q)$
21:         /* Evolution & Evaluation         */
22:         $h_{\text{new}} \leftarrow \text{EVOLVER}(Parents, p_o, C_{\exp})$
23:         Eval $h_{\text{new}}$ on $\mathcal{I}$
24:         /* Experience Management
              (Sec. 3.3)                      */
25:         $e, is\_pos \leftarrow \text{REFLECTOR}(Parents, h_{\text{new}}, F)$
26:         **if** $is\_pos$ **then**
27:            $\mathcal{E}^+ \leftarrow \mathcal{E}^+ \cup \{e\}$
28:         **else**
29:            $\mathcal{E}^- \leftarrow \mathcal{E}^- \cup \{e\}$
30:         **end if**
31:         $\text{UPDATEUTILITY}(C_{\exp}, Parents, h_{\text{new}})$
32:         $\mathcal{E}^+, \mathcal{E}^- \leftarrow \text{PRUNE}(\mathcal{E}^+, \mathcal{E}^-)$
33:         Update statistics for operator $o$
34:         $\mathcal{P}'_t \leftarrow \mathcal{P}'_t \cup \{h_{\text{new}}\}$
35:      **end for**
36:   **end for**
37:   $\mathcal{P}_{t+1} \leftarrow \text{SELECT}(\mathcal{P}_t \cup \mathcal{P}'_t, N)$
38: **end for**
39: **Return** $h^* = \arg\max_{h \in \mathcal{P}_{T_{\max}}} F(h)$

---

*Table 1.* Constructive heuristics on synthetic TSP and 0–1 KP across scales. Each entry is averaged over the same 64 test instances per size. **Bold** indicates the best result; "–" means the heuristic failed to return a valid solution.

| Methods | TSP (Obj. ↓) | | | | | KP (Obj. ↑) | | | | |
|---|---|---|---|---|---|---|---|---|---|---|
| | $n{=}50$ | $n{=}100$ | $n{=}200$ | $n{=}500$ | $n{=}1000$ | $n{=}50$ | $n{=}100$ | $n{=}200$ | $n{=}500$ | $n{=}1000$ |
| Greedy | 6.944 | 9.651 | 13.472 | 20.811 | 28.897 | 20.002 | 40.652 | 57.831 | 90.588 | 128.449 |
| POMO | **5.741** | **7.850** | 12.185 | 22.115 | 35.219 | 19.159 | 39.314 | 56.904 | 87.837 | 128.378 |
| FunSearch | 6.306 | 8.839 | 12.420 | 19.501 | 27.156 | 20.001 | 40.655 | 57.837 | 90.593 | 128.546 |
| EoH | 6.319 | 8.821 | 12.311 | 19.514 | 27.101 | 20.002 | 40.652 | 57.831 | 90.683 | – |
| ReEvo | 6.239 | 8.771 | 12.075 | 18.965 | 26.731 | 20.012 | 40.659 | 57.840 | 90.684 | 128.546 |
| HSEvo | 6.271 | 8.735 | 12.209 | 19.269 | 27.310 | **20.020** | 40.659 | 57.835 | 90.515 | 128.590 |
| MCTS-AHD | 6.214 | 8.732 | 12.247 | 19.194 | 26.717 | 20.007 | 40.655 | 57.830 | 90.683 | 128.842 |
| PartEvo | 6.221 | 8.701 | 12.137 | 18.854 | 26.632 | 20.002 | 40.652 | 57.831 | **90.688** | 128.845 |
| OpenEvolve | 6.204 | 8.639 | 12.074 | 18.792 | 26.211 | 20.013 | 40.663 | 57.843 | 90.682 | 128.836 |
| **RefineEvo** | 6.176 | 8.541 | **11.868** | **18.476** | **25.851** | 20.014 | **40.670** | **57.847** | **90.688** | **128.847** |

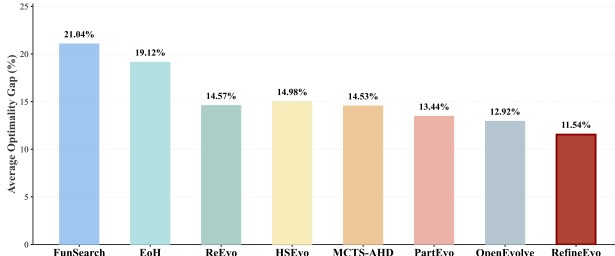

*Figure 2.* Generalization on TSPLIB. The bars represent the mean optimality gap for each method.

# 4. Experiments

## 4.1. Experimental Setup

**Benchmarks and Datasets.** We adopt standard evaluation protocols with distinct settings for evolution and testing to ensure robustness. Detailed configurations are provided in Appendix B.

- *Routing Problems:* We focus on the Traveling Salesperson Problem (TSP) (Matai et al., 2010) and the Capacitated Vehicle Routing Problem (CVRP) (Toth & Vigo, 2002). TSP aims to minimize the total tour length under visitation constraints, while CVRP further introduces vehicle capacity constraints across multiple routes. We additionally include the Vehicle Routing Problem with Time Windows (VRPTW) (Toth & Vigo, 2002), which extends CVRP with time-window feasibility requirements, as a supplementary constructive routing benchmark. Following (Liu et al., 2024b), we sample node coordinates from the unit square $[0, 1]^2$ (Kool et al., 2018) and generate the corresponding demands, capacities, and time windows as specified in Appendix D. We evaluate heuristics on synthetic Euclidean instances across varying scales. For TSP, we also test on real-world TSPLIB (Reinelt, 1991) instances for out-of-distribution evaluation. For construc-

tive routing instances, we use the negative solution cost as fitness; for TSPLIB evaluation, we use the negative average optimality gap.

- *Online Bin Packing Problem (BPP)* (Seiden, 2002): The objective is to assign a stream of $L$ items with sizes $s_i$ to a minimum number of bins with capacity $C$. We generate instances following a Weibull distribution (Castiñeiras et al., 2012) as in (Liu et al., 2024b), evaluating robustness across varying stream lengths and capacity settings. Fitness measures the average lower-bound ratio (Martello & Toth, 1990b).

- *0-1 Knapsack Problem (KP)* (Martello & Toth, 1990a): The objective is to select a subset of $N$ items with values $v_i$ and weights $w_i$ to maximize total value under capacity $W$. Following (Zheng et al., 2025), we generate instances with sizes $N \in \{50, 100, 200\}$ and extend the evaluation to larger scales of 500 and 1000. Fitness represents the total value of the selected items.

- *Multiple Knapsack Problem (MKP)* (Martello & Toth, 1990a): This problem generalizes KP by introducing multiple resource constraints (dimensions) and capacities. We follow the setting in (Ye et al., 2024) to evaluate the evolved ACO heuristics, where fitness corresponds to the total value satisfying all constraints.

**Heuristic Settings.** We apply RefineEvo to evolve high-leverage components within three standard frameworks (formulations in Appendix C). For Constructive Heuristics on TSP, KP, CVRP, VRPTW, and Online BPP, the system evolves the scoring function $H(\cdot)$ to prioritize candidate moves step-by-step. In Ant Colony Optimization (ACO) (Dorigo et al., 2006) for TSP, CVRP, and MKP, RefineEvo designs the heuristic information matrix $\eta$, transforming static distances into dynamic, geometry-aware guidance. For Guided Local Search (GLS) (Arnold & Sörensen, 2019) on TSP, the system evolves the perturbation logic to dynamically penalize solution features and escape local optima.

*Table 2.* Results on Online BPP. The metric is the excess-bin ratio (lower is better) averaged over 64 test streams.

| Methods | 5k_C100 | 10k_C100 | 100k_C100 | 5k_C500 | 10k_C500 | 100k_C500 |
|---|---|---|---|---|---|---|
| First Fit | 2.40% | 2.29% | 2.14% | 0.50% | 0.50% | 0.44% |
| Best Fit | 2.26% | 2.18% | 2.06% | 0.50% | 0.45% | 0.42% |
| FunSearch | 2.25% | 2.20% | 2.06% | 0.10% | **0.05%** | 0.01% |
| EoH | 0.39% | 0.22% | 0.02% | 0.10% | **0.05%** | 0.01% |
| ReEvo | 0.71% | 0.25% | 0.04% | 0.10% | **0.05%** | 0.01% |
| LLM-LNS | 0.54% | 0.33% | 0.03% | 0.99% | 0.47% | 0.04% |
| HSEvo | 0.33% | 0.20% | 0.02% | 0.50% | 0.45% | 0.42% |
| MCTS-AHD | 0.31% | 0.20% | 0.02% | 0.50% | 0.45% | 0.42% |
| **RefineEvo** | **0.25%** | **0.19%** | **0.01%** | **0.05%** | **0.05%** | **0.00%** |

*Table 3.* Constructive heuristics on CVRP and VRPTW.

| Methods | CVRP (Obj. ↓) | | | | VRPTW (Obj. ↓) | | | |
|---|---|---|---|---|---|---|---|---|
| | $n$=50 | $n$=100 | $n$=200 | $n$=500 | $n$=20 | $n$=50 | $n$=100 | $n$=200 |
| PartEvo | 13.393 | 23.226 | 40.307 | 88.096 | 10.828 | 20.471 | 35.157 | 58.489 |
| OpenEvolve | 12.837 | 22.311 | 39.493 | 88.086 | 10.054 | 19.339 | 33.226 | 56.183 |
| **RefineEvo** | **12.385** | **21.630** | **38.041** | **84.601** | **9.862** | **19.018** | **32.628** | **54.746** |

**Baselines and Implementation.** We compare RefineEvo against three categories: (a) Handcrafted Heuristics (e.g., Greedy (Rosenkrantz et al., 1977), Best Fit (Johnson et al., 1974), vanilla ACO/GLS); (b) Neural Combinatorial Optimization (NCO) methods (e.g., POMO (Kwon et al., 2020), DeepACO (Ye et al., 2023)); and (c) LLM-based AHD frameworks (FunSearch (Romera-Paredes et al., 2024), EoH (Liu et al., 2024b), ReEvo (Ye et al., 2024), LLM-LNS (Ye et al., 2025), MCTS-AHD (Zheng et al., 2025), PartEvo (Hu & Zhang, 2026), and OpenEvolve (Sharma, 2025)). All experiments use DeepSeek-v3 (Liu et al., 2024a) as the backbone LLM with a sampling temperature of 1.0. We test robustness to the LLM backbone by using gpt-4o-mini (OpenAI, 2024) and gpt-4o (Hurst et al., 2024) for RefineEvo and all LLM-based AHD baselines. Furthermore, we employ the Qwen text-embedding-v4 model (Zhang et al., 2025b) with an embedding dimension of 1024 to generate embeddings for experience retrieval. We initialize our operator library using exploration and exploitation prompts adapted from EoH. Crucially, RefineEvo dynamically rewrites these descriptions during evolution (see Appendix E.2). Appendix D provides hyperparameters and implementation details. Our code is publicly available at `https://github.com/samwu-learn/RefineEvo`.

### 4.2. Main Results

**Constructive Heuristics.** The constructive heuristic paradigm incrementally builds a feasible solution by making a sequence of local decisions (e.g., selecting the next city in TSP or packing the next item in BPP). We task RefineEvo with evolving the core scoring function that prioritizes these moves, to evaluate the scalability of the generated logic across varying problem scales.

*TSP and KP.* Table 1 reports the performance on synthetic instances. On small-scale TSP ($n \leq 100$), the NCO method POMO retains an advantage, aligning with its strength in fitting fixed training distributions. However, a decisive shift occurs as the problem scale expands ($n \geq 200$). RefineEvo consistently achieves the best objective values, outperforming parameter-heavy neural models and evolution-

ary prompts alike. For instance, at $n = 500$, RefineEvo achieves 18.48, outperforming all LLM-based baselines including the closest competitor OpenEvolve (18.79). This trend confirms that RefineEvo discovers algorithmic logic that scales effectively. A similar pattern emerges on KP: RefineEvo secures the highest objective value starting from $n = 100$ and maintains monotonic improvements on high-dimensional instances ($n = 1000$), demonstrating superior stability where other methods plateau.

*Generalization to Real-World Graphs.* To validate robustness beyond synthetic distributions, we evaluate the best-discovered heuristics on the heterogeneous TSPLIB benchmark. RefineEvo achieves the lowest mean optimality gap of 11.54%, outperforming the next-best method OpenEvolve (12.92%) by a clear margin. This significant performance disparity suggests that RefineEvo's planner-guided mechanism abstracts more generalizable geometric rules, reducing the transfer gap observed in prior methods.

*Online BPP.* The online BPP tests decision-making under strict sequential constraints. As detailed in Table 2, RefineEvo consistently achieves the lowest excess-bin ratio across all settings. A notable divergence appears in the large-capacity regimes (e.g., 5k_C500), where the decision space expands significantly. In these settings, recent methods like MCTS-AHD and HSEvo degrade to excess-bin ratios of 0.42%–0.50%, which are comparable to the Best Fit baseline. In contrast, FunSearch and EoH remain more stable (0.01%–0.10%), yet RefineEvo further pushes the boundary of precision, achieving an excess-bin ratio of 0.00% on the largest scale. This underscores that while some AHD methods falter under expanded decision spaces, RefineEvo's experience-driven evolution sustains precise exploration, delivering the most robust solution quality.

*CVRP and VRPTW.* We also evaluate RefineEvo on constrained routing problems, CVRP and VRPTW, comparing against PartEvo and OpenEvolve (Table 3). RefineEvo achieves the best objective values across all scales on both tasks, with larger absolute gains on larger instances. For example, on CVRP500, RefineEvo achieves 84.601, compared with 88.086 from OpenEvolve and 88.096 from PartEvo. A similar trend holds on VRPTW, where RefineEvo reduces total cost from 56.183 (OpenEvolve) to 54.746 at $n = 200$. These results suggest that RefineEvo's planning-guided and experience-driven evolution can extend to routing problems with richer feasibility constraints.

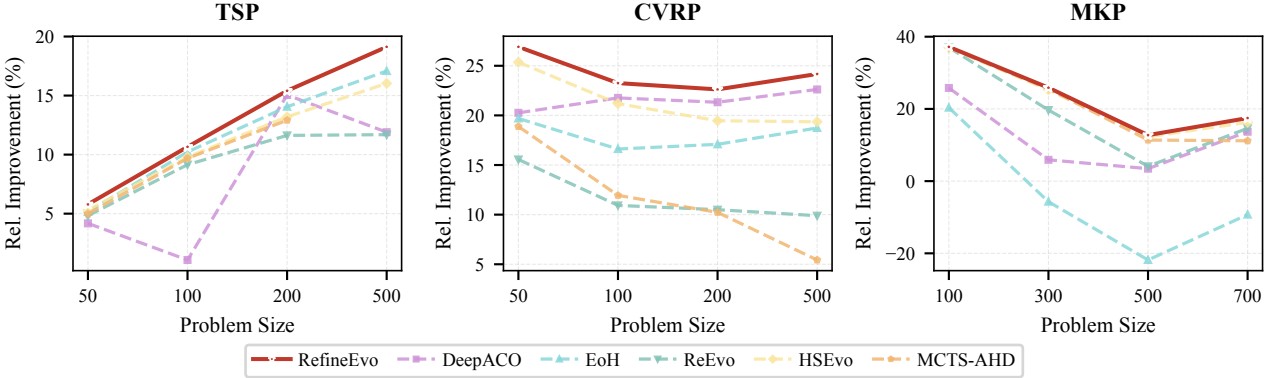

*Figure 3.* Relative improvement (%) of evolved heuristics over the vanilla ACO across varying problem sizes on TSP, CVRP, and MKP.

*Table 4.* Performance of GLS on TSP. The table reports the average optimality gap and runtime per instance (lower is better for both).

| Methods | TSP50 | | TSP100 | | TSP200 | |
|---|---|---|---|---|---|---|
| | gap (%) | time (ms) | gap (%) | time (ms) | gap (%) | time (ms) |
| GLS | 0.0169 | 21.5 | 0.0023 | 1014.3 | 0.2840 | 1570.8 |
| EoH | 0.0041 | 34.6 | 0.0035 | 1076.9 | 0.4368 | 1785.9 |
| ReEvo | **0.0000** | 22.5 | 0.0101 | 1041.2 | 0.3045 | 1617.0 |
| LLM-LNS | **0.0000** | 39.1 | 0.0475 | 1051.8 | 0.3873 | 1590.1 |
| MCTS-AHD | **0.0000** | 18.2 | 0.0006 | 849.2 | 0.2339 | 1331.9 |
| **RefineEvo** | **0.0000** | **17.0** | **0.0000** | **843.7** | **0.2240** | **1326.8** |

We additionally report mean and standard deviation over three independent runs in Appendix F.

*Table 5.* Planner ablations. Random Mode randomly chooses exploration or exploitation. Random/Planner Selection picks exactly $k$ operators. Fixed-Interval Refinement updates each operator individually when it shows no improvement for $k$ generations.

| Methods | TSPLIB ↓ | TSP100 ↓ | BPP 5k_C100 ↓ |
|---|---|---|---|
| **RefineEvo** | **11.54%** | 8.541 | **0.25%** |
| Select All | 11.63% | **8.496** | **0.25%** |
| Random Mode | 14.74% | 8.665 | 0.27% |
| Random Selection (best $k$=4) | 15.14% | 8.639 | 0.30% |
| Planner Selection (best $k$=4) | 13.15% | 8.603 | 0.27% |
| w/o Refinement | 15.63% | 8.702 | 0.30% |
| Fixed-Interval Refinement (best $k$=3) | 14.05% | 8.651 | 0.28% |
| Select All & w/o Refinement | 14.76% | 8.632 | 0.27% |

**Ant Colony Optimization** We further apply RefineEvo to optimize the heuristic information component ($\eta$) within the ACO framework, replacing the static handcrafted terms (e.g., inverse distance) typically used to guide transition probabilities. RefineEvo evolves expressive formulas that dynamically leverage node geometry and pheromone trails, requiring no additional training. Figure 3 summarizes the relative improvement across problem sizes on TSP, CVRP, and MKP. Specifically, RefineEvo exhibits superior scalability on TSP as its relative improvement grows linearly with problem size to reach nearly 20% at $N = 500$. This trend significantly widens the gap against DeepACO. Furthermore, RefineEvo maintains a $> 20\%$ gain on CVRP across all scales while baselines such as MCTS-AHD degrade noticeably as complexity increases. On MKP, while RefineEvo remains competitive with strong AHD baselines at smaller scales, its advantage widens significantly as problem complexity increases. This suggests that RefineEvo's experience pool captures transferable geometric patterns that static inverse-distance heuristics fail to exploit.

Additional CVRPLib results are provided in Appendix G.6, where RefineEvo further reduces the ACO gap from 21.13% to 16.91%.

**Guided Local Search** Guided Local Search escapes local optima by iteratively penalizing undesirable solution features (e.g., long edges in TSP) and re-optimizing under the augmented objective. The core challenge lies in designing a utility update rule that determines which features to penalize and how strongly to perturb the landscape. We task RefineEvo with designing this edge-utility adjustment logic. As reported in Table 4, RefineEvo achieves the best (or tied-best) optimality gaps while simultaneously reducing the computational runtime. For instance, on TSP100, RefineEvo achieves 0.00% gap in 843.7ms, compared to MCTS-AHD's 0.0006% in 849.2ms. This dual advantage indicates that the evolved update rule achieves more effective perturbations, suggesting that the learned logic better balances exploitation of current structure with exploration of alternative configurations to escape local optima faster while maintaining solution quality.

Collectively, the results across these paradigms demonstrate that RefineEvo consistently discovers superior heuristics regardless of the underlying search framework.

### 4.3. Ablation Studies

We validate the Planner and Bidirectional Experience Pool components on TSPLIB, TSP100, and Online BPP.

**Ablation on Planner**    We isolate operator selection and operator refinement.

*Operator Selection.*    As shown in Table 5, RefineEvo (11.54%) slightly outperforms Select All (11.63%), which applies all operators indiscriminately. Although the performance margin is modest on TSPLIB, the efficiency gain is substantial, since RefineEvo consumes only ∼511k tokens compared to ∼965k for the Select All baseline. This demonstrates that selective scheduling yields a nearly twofold improvement in token efficiency. We hypothesize that as the population matures, certain operators become redundant or counterproductive, and the Planner learns to suppress these dynamically. To further assess decision quality, we compare against stochastic baselines, such as Random Mode (14.74%). We also evaluate fixed-budget selection, where exactly $k$ operators are chosen per generation. As detailed in Appendix G, both methods achieve their peak performance at $k = 4$. However, at this optimal setting, Planner Selection significantly outperforms Random Selection across all metrics: 13.15% vs 15.14% on TSPLIB, and 0.27% vs 0.30% on BPP. This confirms that with sufficient budget, the Planner identifies synergistic operator subsets by reasoning about the current search state, a capability that random methods inherently lack.

*Operator Refinement.*    Disabling refinement degrades TSPLIB performance from 11.54% to 15.63% (Table 5), demonstrating that static prompts cannot sustain progress. To visualize this effect, Appendix E.3 shows how refinement revives underperforming operators. We further compare against Fixed-Interval Refinement, which updates operators that fail to yield improvement for $k$ consecutive generations. The best setting ($k$=3) yields 14.05%, which is still inferior to RefineEvo (see Appendix G for results with other $k$ values), highlighting that rigid schedules lack the adaptability of event-triggered diagnosis. Similar trends hold on other tasks where disabling refinement increases the BPP ratio from 0.25% to 0.30%. Finally, the Select All & w/o Refinement baseline yields 14.76%, which represents a significant degradation compared to RefineEvo (11.54%), confirming that even when all operators are available, the lack of evolutionary refinement leads to suboptimal performance.

**Ablation on Bidirectional Experience Pool**    We analyze three core components including bidirectional feedback, trajectory tracking, and situational grounding (Table 6). Removing the pool entirely (w/o Experience) causes the largest drop (15.91%), confirming its necessity. Disaggregating feedback reveals that removing negative experiences (13.94%) degrades performance more than removing positive ones (12.92%), suggesting that explicit warnings against pitfalls are critical. Replacing parent-to-offspring trajectories with population comparisons (w/o trajectory) widens the gap to 15.32%, indicating that causal lineage

*Table 6.* BEP ablations. w/o trajectory replaces parent-to-offspring tracking with population pairwise comparisons; w/o situation removes retrieval constraints, injecting all past experiences indiscriminately.

| Methods | TSPLIB ↓ | TSP100 ↓ | BPP 5k_C100 ↓ |
|---|---|---|---|
| RefineEvo | 11.54% | 8.541 | 0.25% |
| w/o Experience | 15.91% | 8.687 | 0.38% |
| w/o Negative Experience | 13.94% | 8.662 | 0.34% |
| w/o Positive Experience | 12.92% | 8.640 | 0.29% |
| w/o situation | 14.33% | 8.635 | 0.33% |
| w/o trajectory | 15.32% | 8.642 | 0.36% |
| w/o trajectory & situation | 15.51% | 8.645 | 0.37% |

*Table 7.* Cross-task universality. Performance on Target Tasks (rows) uses experience pools initialized from different Source Tasks (columns). "None" denotes no experience provided. To ensure fair comparison, all methods disable new experience accumulation, relying solely on the provided source pool. **Bold** indicates the best result.

| Target Task | Scale | None | Source of Experience | | |
|---|---|---|---|---|---|
| | | | TSP | KP | BPP |
| TSP Obj ↓ | $n = 100$ | 8.687 | **8.602** | 8.813 | 8.657 |
| | $n = 500$ | 18.997 | **18.686** | 19.294 | 18.832 |
| KP Obj ↑ | $n = 100$ | 40.659 | 40.666 | **40.668** | 40.660 |
| | $n = 500$ | 90.677 | 90.675 | **90.684** | 90.679 |
| BPP Ratio ↓ | 5k items | 0.38% | 0.28% | 0.31% | **0.25%** |
| | 10k items | 0.25% | **0.20%** | 0.25% | **0.20%** |

provides more actionable signals. Finally, disabling situational retrieval (w/o situation) yields 14.33%, confirming that context-aware filtering prevents the misapplication of mismatched strategies or irrelevant pitfalls.

To further isolate the role of contextual matching in the BEP, we compare semantic retrieval with two simpler alternatives (Table 8). Random samples experiences uniformly from the pool, while TF–IDF ranks experiences by lexical term overlap with the current query. Among the three, Random performs worst, indicating that injecting irrelevant experiences can introduce noisy guidance. TF–IDF improves over Random by capturing surface keyword similarity, yet it may still retrieve experiences with similar terms but different applicability conditions or even opposite recommendations. By contrast, semantic retrieval achieves the best or tied-best performance across all settings, with clearer gains on larger TSP instances (e.g., 18.476 vs 19.081 and 19.528 on TSP500). These results confirm that effective experience reuse requires matching not only lexical content, but also the contextual applicability of past experiences.

**Cross-Task Universality**    We examine transferability by initializing the BEP using final experience pools from different source tasks. To ensure a strictly fair comparison, these experiments disable the accumulation of new experiences and force the model to rely solely on the provided

*Table 8.* Ablation on experience retrieval strategies. Semantic retrieval is compared with Random and TF–IDF alternatives.

| Strategy | TSP (Obj. ↓) | | | BPP (Ratio ↓) | | |
|---|---|---|---|---|---|---|
| | $n=100$ | $n=200$ | $n=500$ | 5k_C100 | 10k_C100 | 100k_C100 |
| Random | 8.742 | 12.247 | 19.528 | 0.43% | 0.28% | 0.03% |
| TF–IDF | 8.657 | 12.107 | 19.081 | 0.36% | 0.20% | **0.01%** |
| Semantic | **8.541** | **11.868** | **18.476** | **0.25%** | **0.19%** | **0.01%** |

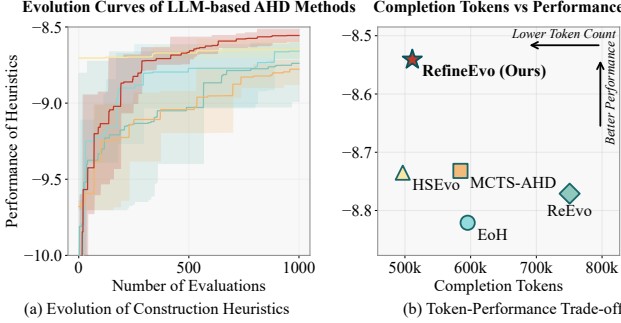

**Evolution Curves of LLM-based AHD Methods** **Completion Tokens vs Performance**

(a) Evolution of Construction Heuristics

(b) Token-Performance Trade-off

*Figure 4.* Efficiency analysis of Construction Heuristics for TSP. (a) Evolution of the optimality gap over evaluations. (b) Trade-off between final performance and computational cost (token usage).

source pool. As detailed in Table 7, pre-evolved sources consistently outperform the zero-shot baseline where no experience is provided. Results show that in-domain transfer typically yields the largest gains. However, TSP derived experience transfers effectively to BPP, reducing the ratio from 0.38% to 0.28% on the 5k dataset. This implies that the routing strategies learned in TSP share underlying structural similarities with bin packing logic, allowing for effective cross-domain generalization. We further analyze BEP retrieval overhead in Appendix G.4.

**Efficiency Analysis** We evaluate efficiency regarding convergence speed and token cost as visualized in Figure 4. The evolution curve in (a) shows that RefineEvo improves rapidly under a fixed evaluation budget. The cost-performance profile in (b) plots the final objective value against cumulative token usage. RefineEvo achieves the best performance while consuming fewer tokens than competitive baselines. Despite slightly higher cost than HSEvo, it yields significantly better solution quality, establishing a superior Pareto frontier in the trade-off between performance and computational cost. The detailed token breakdown per method is provided in Appendix G.5.

**Backbone Robustness** To evaluate the robustness of RefineEvo to different LLM backbones, we instantiate RefineEvo and representative baselines under three LLM backbones of varying capability: GPT-4o-mini (OpenAI, 2024), DeepSeek-V3 (Liu et al., 2024a) (default), and GPT-4o (Hurst et al., 2024). As shown in Table 9, RefineEvo achieves the best performance across all reported backbones and metrics. Even with the smaller GPT-4o-mini

*Table 9.* Backbone robustness on constructive TSP and Online BPP. Bold indicates the best result under each backbone and metric.

| Backbone | Method | TSP100 ↓ | TSP500 ↓ | BPP 5k_C100 ↓ |
|---|---|---|---|---|
| GPT-4o-mini | EoH | 8.841 | 19.317 | 0.63% |
| | ReEvo | 8.821 | 19.189 | 0.73% |
| | **RefineEvo** | **8.712** | **19.101** | **0.36%** |
| DeepSeek-V3 | EoH | 8.821 | 19.514 | 0.39% |
| | ReEvo | 8.771 | 18.965 | 0.71% |
| | **RefineEvo** | **8.541** | **18.476** | **0.25%** |
| GPT-4o | EoH | 8.724 | 19.129 | 0.48% |
| | ReEvo | 8.569 | 18.584 | 0.71% |
| | **RefineEvo** | **8.534** | **18.503** | **0.43%** |

backbone, RefineEvo (8.712 on TSP100) outperforms both EoH (8.841) and ReEvo (8.821) under the same backbone, suggesting that the planning-guided and experience-driven mechanisms remain effective across different models. Additional results are provided in Appendix G.2.

## 5. Conclusion

This paper proposes RefineEvo, a multi-agent evolutionary framework for automated heuristic design using LLMs. By integrating a Planner for state-aware operator scheduling and a Bidirectional Experience Pool that leverages both positive and negative insights, RefineEvo establishes a robust mechanism for autonomous discovery and continuous improvement. We have evaluated the framework across a diverse set of combinatorial optimization problems and algorithmic paradigms. Experiments demonstrate that RefineEvo consistently outperforms state-of-the-art baselines while maintaining high computational efficiency. This suggests that coordinating specialized agents with experience-guided evolution provides a practical and scalable solution for heuristic design. Future work includes extending RefineEvo to constrained optimization and multi-objective settings, as well as investigating theoretical foundations of experience-driven search dynamics.

## Impact Statement

This paper presents work aiming to advance the field of Machine Learning. We identify specific limitations and potential risks associated with our approach.

**Limitations.** Our evaluation primarily focuses on classic combinatorial optimization benchmarks. Consequently the generalization to real-world industrial problems characterized by complex constraints remains to be validated. Furthermore the performance relies on the code generation capabilities of the underlying LLM, implying that results may vary across different model families. Additionally we currently lack a theoretical analysis regarding the convergence properties of evolutionary search guided by experi-

ence.

**Broader Impacts.** RefineEvo aims to significantly reduce manual effort in algorithm design. However automated code generation carries inherent risks of producing incorrect or inefficient solutions if deployed without proper verification. We strongly recommend maintaining human oversight before applying evolved heuristics in safety critical domains.

## Acknowledgements

This work was supported in part by the Strategic Priority Research Program of Chinese Academy of Sciences (XDA0480203), the National Key R&D Program of China (No. 2025ZD0122000), NSFC 62273347, the Key Research and Development Program of Jiangsu Province (BE2023016).

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

## A. Comparison with Existing Frameworks

*Table 10.* Functional comparison of representative LLM-based AHD frameworks.

| Method | population-based evolutionary | thought & code evolution | reflection or experience | explicit explore–exploit | operator Refinement |
|---|:---:|:---:|:---:|:---:|:---:|
| FunSearch | ✓ | ✗ | ✗ | ✗ | ✗ |
| EoH | ✓ | ✓ | ✗ | ✗ | ✗ |
| ReEvo | ✓ | ✗ | ✓ | ✗ | ✗ |
| HSEvo | ✓ | ✗ | ✓ | ✗ | ✗ |
| MCTS-AHD | ✗ | ✓ | ✗ | ✓ | ✗ |
| RefineEvo | ✓ | ✓ | ✓ | ✓ | ✓ |

## B. Definition of Tasks

We consider a set of NP-hard combinatorial optimization (CO) problems. Each task specifies (i) an instance space (e.g., distances, coordinates, demands, item values/weights), (ii) a solution representation (e.g., tours, routes, subsets, assignments), and (iii) an objective function to minimize (routing/grouping) or maximize (subset selection).

### B.1. CO Problems

**Traveling Salesman Problem** The Traveling Salesman Problem (TSP) aims to find the shortest Hamiltonian cycle that visits each city exactly once and returns to the start. Given an instance with $N$ nodes and a distance (cost) matrix $D = \{d_{j,k}\}_{j,k=1}^{N}$, a solution is a permutation $s = (s_1, \ldots, s_N)$ of node indices. The objective is

$$\min_{s} \quad f(s) = \sum_{t=1}^{N-1} d_{s_t, s_{t+1}} + d_{s_N, s_1}, \tag{2}$$

subject to visiting each node exactly once (equivalently, each node has degree two in the resulting tour and no subtours exist).

**Bin Packing Problem** The Bin Packing Problem (BPP) requires packing items of different sizes into bins of fixed capacity $W$ while minimizing the number of bins used. A standard formulation uses assignment variables $x_{i,b} \in \{0,1\}$ (item $i$ placed in bin $b$) and bin-usage variables $y_b \in \{0,1\}$:

$$\min \quad \sum_{b} y_b \tag{3}$$
$$\text{s.t.} \quad \sum_{b} x_{i,b} = 1, \quad \forall i,$$
$$\sum_{i} a_i x_{i,b} \leq W \cdot y_b, \quad \forall b,$$
$$x_{i,b}, y_b \in \{0,1\},$$

where $a_i$ denotes the size of item $i$. In the *online* setting, items arrive sequentially and the algorithm must irrevocably decide a bin for each item upon arrival, while *offline* BPP assumes all items are known beforehand.

**0-1 Knapsack Problem** The 0-1 Knapsack Problem (KP) selects a subset of items to maximize total value subject to a capacity constraint, where each item can be chosen at most once. Given item values $\{v_i\}_{i=1}^{N}$, weights $\{w_i\}_{i=1}^{N}$, and knapsack capacity $W$, the canonical formulation is

$$\max \quad \sum_{i=1}^{N} v_i x_i \tag{4}$$
$$\text{s.t.} \quad \sum_{i=1}^{N} w_i x_i \leq W, \quad x_i \in \{0,1\} \, \forall i,$$

where $x_i = 1$ indicates selecting item $i$.

**Multiple Knapsack Problem** The Multiple Knapsack Problem (MKP) typically refers to distributing items among multiple knapsacks. However, in many ACO benchmarks, MKP is instantiated as the *Multidimensional Knapsack Problem*, which selects a subset of items under multiple resource constraints (dimensions). A common formulation is:

$$\max \quad \sum_{i=1}^{N} v_i x_i \tag{5}$$

$$\text{s.t.} \quad \sum_{i=1}^{N} w_{i,k} x_i \leq C_k, \quad k = 1, \ldots, m, \quad x_i \in \{0, 1\} \ \forall i,$$

where $w_{i,k}$ is the weight of item $i$ under constraint (dimension) $k$, and $C_k$ is the corresponding capacity.

**Capacitated Vehicle Routing Problem** The Capacitated Vehicle Routing Problem (CVRP) extends routing by imposing vehicle capacity limits: vehicles start and end at a depot, serve customer demands, and the goal is to minimize total travel distance. An instance contains a depot (node 0), customers $\{1, \ldots, N\}$, a distance matrix $D = \{d_{i,j}\}$, customer demands $\{\delta_i\}_{i=1}^{N}$, and vehicle capacity $C$. A solution $s = \{\rho_1, \ldots, \rho_q\}$ consists of $q$ routes. Let $V(\rho_j)$ denote the set of customers served by route $\rho_j$, and let the sequence of route $\rho_j$ be $(0, v_1, v_2, \ldots, v_m, 0)$. The problem is formulated as:

$$\min_{s} \quad \sum_{j=1}^{q} \text{Length}(\rho_j) \tag{6}$$

$$\text{s.t.} \quad \sum_{i \in V(\rho_j)} \delta_i \leq C, \quad \forall j = 1, \ldots, q, \tag{7}$$

$$\bigcup_{j=1}^{q} V(\rho_j) = \{1, \ldots, N\}, \tag{8}$$

$$V(\rho_j) \cap V(\rho_k) = \emptyset, \quad \forall j \neq k, \tag{9}$$

where $\text{Length}(\rho_j) = d_{0,v_1} + \sum_{t=1}^{m-1} d_{v_t,v_{t+1}} + d_{v_m,0}$ represents the total round-trip distance of route $\rho_j$. The constraints ensure that no vehicle exceeds its capacity $C$ and that every customer is visited exactly once.

## C. Paradigms

**Constructive Heuristics** Constructive Heuristics builds a feasible solution incrementally. Starting from an empty state, the algorithm repeatedly selects a component to add to the partial solution until the task is completed. Let $S_t$ be the partial solution at step $t$, and $\mathcal{A}(S_t)$ be the set of available candidate actions (e.g., unvisited cities in TSP, unpacked items in BPP). The decision is typically made by a scoring function $H(S_t, a)$:

$$a^* = \underset{a \in \mathcal{A}(S_t)}{\arg\max} \ H(S_t, a). \tag{10}$$

- **Routing (TSP):** The action $a$ corresponds to the next node to visit. The scoring function $H$ usually involves the distance $d_{current,a}$.

- **Packing (BPP):** In the online setting, for an arriving item $i$, the action $a$ is the choice of bin $b$. $H$ balances fit quality (e.g., Best Fit) and capacity preservation.

- **Subset Selection (KP):** The greedy strategy often sorts items by a density ratio (e.g., $v_i/w_i$). Here, $H$ is static or dynamically updated based on remaining slack in multiple dimensions.

LLM-based methods (AHD) often focus on learning or evolving the mathematical form of $H(\cdot)$ to outperform standard human-designed greedy rules.

**Ant Colony Optimization**    Ant Colony Optimization (ACO) is a population-based metaheuristic where agents (ants) construct solutions probabilistically. The decision to select component $j$ after component $i$ is governed by a pheromone matrix $\tau$ (learned historical desirability) and a heuristic information matrix $\eta$ (problem-specific immediate desirability). The probability is typically given by:

$$P(j|i) = \frac{[\tau_{ij}]^\alpha [\eta_{ij}]^\beta}{\sum_{k \in \mathcal{N}_i} [\tau_{ik}]^\alpha [\eta_{ik}]^\beta}, \tag{11}$$

where $\mathcal{N}_i$ is the set of feasible neighbors.

- **TSP/CVRP:** $\tau_{ij}$ and $\eta_{ij}$ are defined on edges. A classic heuristic is $\eta_{ij} = 1/d_{ij}$. For CVRP, $\eta$ may also integrate savings values or demand information relative to capacity.

- **MKP:** The construction is often node-based rather than edge-based. $\tau_j$ represents the desirability of selecting item $j$, and the heuristic value is typically derived from item utility, e.g., $\eta_j = v_j/(\sum_k w_{j,k}/C_k)$.

In modern AHD frameworks, the LLM functions as a higher-level designer that generates the code for calculating $\eta$ based on the specific constraints (distances, demands, weights) of the target problem.

**Guided Local Search**    Guided Local Search (GLS) improves a complete solution $s$ by iteratively applying local operators (e.g., 2-opt, Swap, Relocate) to reach a local optimum, combined with a penalty mechanism to escape it. GLS augments the original objective function $f(s)$ with a penalty term based on solution features (e.g., long edges in TSP). The augmented objective $g(s)$ is:

$$g(s) = f(s) + \lambda \sum_i p_i \cdot I_i(s), \tag{12}$$

where $I_i(s)$ is an indicator function (e.g., 1 if edge $i$ is present in solution $s$), $p_i$ is the current penalty counter for feature $i$, and $\lambda$ is a regularization parameter.

- **TSP:** Features are typically edges. When stuck in a local optimum, GLS penalizes the long edges present in the current tour ($p_{edge} \leftarrow p_{edge} + 1$), making them "expensive" and encouraging the local search to break them.

- **Knowledge-Guided (KGLS):** Instead of simple penalties, an LLM-generated heuristic can analyze the instance distribution (e.g., clustered cities in CVRP) to define a *knowledge matrix* or custom perturbation moves that guide the search more intelligently than random kicks.

## D. Experiment Details

### D.1. Experimental Setup

This section details the evaluation protocol used in the heuristic evaluation phase of RefineEvo. Following common practice in LLM-based AHD, we specify the *evaluation dataset $\mathcal{I}$* (the set of instances used to compute fitness). An *evaluation* corresponds to executing a generated heuristic on the instances in $\mathcal{I}$ and computing an aggregated fitness score. For constructive heuristics, one run produces a single solution per instance (or an episode for online problems). For metaheuristics (e.g., ACO/GLS), one evaluation runs the algorithm under a fixed inner-loop budget (time limit or iteration limit) and reports the best solution found.

**Constructive heuristics (TSP/KP/Online BPP).**    We instantiate $\mathcal{I}$ differently depending on the task:

- **TSP (constructive):** $\mathcal{I}$ contains a set of small-to-medium Euclidean instances (e.g., 50-node instances), enabling fast rollouts while preserving meaningful structure.

- **KP:** $\mathcal{I}$ contains randomly generated knapsack instances with fixed item counts and capacity ratios (e.g., 100 items with a standard capacity setting), consistent with the main paper.

- **Online BPP:** $\mathcal{I}$ includes instances with multiple item counts (e.g., 1,000 and 5,000) and multiple bin capacities (e.g., $W = 100$ and $W = 500$), so that the evolved policy does not collapse to a single scale.

*Table 11.* Composition of the evaluation dataset $\mathcal{I}$ used during search.

| Paradigm | Task | Evaluation dataset $\mathcal{D}$ |
|---|---|---|
| Constructive Heuristics | TSP | 64 Euclidean TSP instances with $n = 100$ |
| Constructive Heuristics | KP | 64 KP instances with 100 items |
| Constructive Heuristics | Online BPP | 4 WeiBull BPP instances: $(n, W) \in \{(1,000, 100), (1,000, 500), (5,000, 100), (5,000, 500)\}$ |
| ACO | TSP | 5 TSP instances with $n = 50$ |
| ACO | CVRP | 10 CVRP instances with 50 customers |
| ACO | MKP | 5 MKP instances with 100-item (m=5) |
| GLS | TSP | 10 synthetic instances with $n = 100$ |

*Table 12.* Key hyperparameters.

| Item | Value |
|---|---|
| Temperature | 1.0 |
| Population size $N$ | 10 |
| Generations $T$ | 30 |
| Initial population size $N$ | 30 |
| Improvement window | 3 |
| Top-$k$ retrieval | 3 |
| Embedding dimension | 1024 |
| Per-instance time limit | 300 |
| runs for evaluation | 3 |

**ACO (TSP/MKP/CVRP).** For ACO-based heuristic design, each element of $\mathcal{D}$ is an instance on which ACO is run with a fixed inner-loop budget. We keep *all ACO hyperparameters fixed* across methods (number of ants, evaporation rate, iteration budget, candidate list size, etc.) and only allow the heuristic components specified in the main paper (e.g., the scoring rule or adaptive filtering mechanism) to be designed by the LLM.

**GLS (TSP).** For GLS-style heuristics, we similarly fix the metaheuristic shell (penalty update schedule, perturbation scheme, and stopping rule) and allow edits only to the target heuristic components (e.g., augmented cost terms or move selection rules). Each evaluation runs GLS for a fixed inner-loop budget and returns the best-found objective.

**LLM inference settings.** We employed Qwen text-embedding-v4 (Zhang et al., 2025b) as the embedding model, with an embedding dimension of 1024. The exact backbone model used in the default setting are reported in the main paper, and additional backbones are evaluated in Appendix G.1.

**Hyperparameters.** We recommend explicitly listing the parameters of RefineEvo in Table 12. For almost all tasks, each LLM-based AHD baseline is allocated an evaluation budget of 1000 (maximum number of evaluations) on the evaluation set. To mitigate statistical variance, we perform three independent runs per method for each application setting. For EoH, we use 20 generations. For online BPP, we set the population size to 20, resulting in 2020 evaluations. For TSP, we initialize 20 candidates and retain 10 feasible ones as the effective population, resulting in 1020 evaluations.

# E. Detailed Methodology

### E.1. Operator Selection

In each generation $t$, the Planner characterizes the current search dynamics using population diversity $D_t$ and convergence rate $R_t$. We compute diversity directly from fitness statistics as the normalized standard deviation of the population's objective values:

$$D_t = \frac{\text{Std}\big(\{F_i^{(t)}\}_{i=1}^N\big)}{\max\big(|F_{\text{worst}}^{(t)} - F_{\text{best}}^{(t)}|, \epsilon\big)}, \tag{13}$$

where $F_i^{(t)}$ denotes the objective value of individual $i$ in generation $t$, and $\epsilon$ avoids division by zero. This measure increases when the population spans a broader range of fitness values and decreases as it concentrates around similar-quality solutions.

To detect stagnation or convergence, we compute the improvement rate with window size $w$:

$$R_t = \frac{F_{\text{best}}^{(t)} - F_{\text{best}}^{(t-w)}}{\max\big(|F_{\text{best}}^{(t-w)}|, \epsilon\big)}, \tag{14}$$

where $F_{\text{best}}^{(t)}$ is the best objective value in generation $t$. We set the window size $w = 1$ in our experiments.

The Planner outputs a JSON object to make operator scheduling deterministic. It includes (1) the selected strategy type, (2) the chosen operators, (3) a brief justification grounded in the observed metrics, and (4) an overall confidence score used as a lightweight indicator for budget allocation.

**Output for Operator Selection**

```
{
  "strategy": "exploitation",

  "reasoning": "Population shows steady improvement (best objective decreasing
  from 8.742 to 8.674) with increasing diversity (0.272 to 0.331), indicating
  successful exploration that now requires refinement. Low standard deviation
  (0.035) and high improvement potential (0.399) suggest concentrated progress
  around promising solutions.",

  "selected_operators": ["op3", "op4"],

  "confidence": 0.85
}
```

### E.2. Operator Refinement

While operator selection adapts *which* operators to apply, operator refinement adapts *how* each operator guides generation over long horizons. We maintain simple operator-level traces such as recent invocation counts, the fraction of offspring that outperform their best parent, and the distribution of improvements conditioned on each operator. When the global search shows stagnation or when a specific operator consistently fails to generate useful variants, we trigger an operator diagnosis step.

The Improvement Checker decides whether the operator should be refined and how urgently. We distinguish two refinement modes: **incremental** refinement adds precise constraints or emphasizes effective sub-behaviors while preserving the original intent; **radical** refinement reorients the operator intent when the current description is misaligned with the population regime (e.g., over-simplifying when the population benefits from structured hybrid mechanisms), or when the operator repeatedly produces degenerate offspring.

**Output for Improvement Checker**

```
{
  "needs_improvement": true,

  "reasoning": "Operator shows clear degradation with mean fitness worsening
  from ˜9.6 to ˜15.9 across generations, including a severe outlier of 19.8 in
  Gen 13, despite maintaining good success rates. The current population
  context shows low diversity (0.2979) and zero convergence rate, indicating
  stagnation where operator refinement could break local optima.",

  "confidence": 0.8,

  "improvement_urgency": "medium"
}
```

When refinement is triggered, the Operator Refining prompt rewrites the operator description by leveraging population statistics, and salient traits of top-performing heuristics. The rewritten description is designed to be *actionable* , *distinct* from other operators, and *compatible* with efficiency constraints. Table 13 reports the initial and final operator descriptions to design step-by-step construction heuristics for TSP, showing how the operator library can shift from generic novelty/simplification goals to more structured guidance aligned with the discovered effective patterns. Table 14 further illustrates an operator's multi-step evolution, where radical updates reorient intent under misalignment, and incremental updates add precision once the population improves.

*Table 13.* Operator descriptions to design constructive heuristics for TSP before and after refinement.

| Op. | Initial description | Final description |
| --- | --- | --- |
| op1 | create a new algorithm that has a totally different form from the given ones | design novel algorithms that maintain computational efficiency while ensuring consistent exploration beyond local optima. |
| op2 | create a new algorithm that has a totally different form from the given ones but can be motivated from them | explore algorithms that enhance adaptive lookahead with intelligent dynamic candidate filtering, destination-aware threshold optimization, and multi-phase regret-progressive evaluation. |
| op3 | create a new algorithm that has a different form but can be a modified version of the algorithm provided | create a refined algorithm that specifically focuses on dynamic lookahead optimization to escape local optima while maintaining destination-aware filtering. |
| op4 | identify the main algorithm parameters and create a new algorithm that has a different parameter settings of the score function provided | create algorithms that integrate adaptive candidate filtering with sophisticated path-based scoring mechanisms to achieve breakthrough performance improvements. |
| op5 | identify the main components in the function and simplify the components to enhance the generalization to potential out-of-distribution instances | focus on identifying and reinforcing the most effective components from top performers, with special emphasis on maintaining destination-aware path optimization and adaptive lookahead mechanisms that have proven most successful. |

*Table 14.* Refinement of operator `op5` during RefineEvo.

| Gen. | Type | Diagnosis | Updated description of `op5` |
|------|------|-----------|------------------------------|
| 0 (init) | – | Initial operator intent. | Identify the main components in the function and simplify the components to enhance the generalization to potential out-of-distribution instances. |
| 21 | radical | The population has converged on sophisticated hybrid approaches (adaptive lookahead, dynamic thresholding, candidate filtering) that are being lost through excessive simplification. | Identify core algorithmic components and enhance their robustness through adaptive mechanisms and intelligent parameterization rather than simplification, focusing on maintaining performance while improving generalization to out-of-distribution instances. |
| 23 | radical | The population context shows high diversity but poor convergence, suggesting the operator needs a fundamental shift toward more focused, performance-driven refinement rather than open-ended adaptation. | Focus on identifying and reinforcing the most effective algorithmic components from top performers, avoiding unnecessary complexity while maintaining generalization capabilities. |
| 26 | incremental | Overall improving with strong mean/best values, but occasional spikes suggest adding more precise guidance: keep simplicity while preserving proven effective components. | Focus on identifying and reinforcing the most effective components from top performers, with special emphasis on maintaining destination-aware path optimization and adaptive lookahead mechanisms that have proven most successful. |

**Output for Operator Refining**

```
{
  "improvement_type": "incremental",

  "reasoning":"Overall improving with strong mean/best values, but occasional
  spikes suggest adding more precise guidance: keep simplicity while
  preserving proven effective components.",

  "new_description": "Focus on identifying and reinforcing the most effective
  components from top performers, with special emphasis on maintaining
  destination-aware path optimization and adaptive lookahead mechanisms that
  have proven most successful.",

  "confidence": 0.85
}
```

## E.3. Operator Survival Analysis

Figure 5 reveals a clear contrast. Without operator refinement, only a few operators dominate the early generations, while most operators produce almost no surviving offspring afterwards, leading to long stretches of zeros and effectively operator-level stagnation. This suggests that fixed operators become progressively misaligned with the evolving search landscape, and thus fail to contribute in later stages. In contrast, with operator refinement only, operator survival remains substantially more persistent and diverse across generations, where multiple operators repeatedly regain non-trivial survival. Importantly, the radical operator improvement events (circled) are triggered precisely when an operator becomes temporarily unproductive.

## E.4. Bidirectional Experience Pool

RefineEvo maintains a bidirectional experience pool to provide retrieval-augmented guidance to generation. The pool stores two complementary memory banks: (i) successful experiences that summarize reusable patterns associated with improvements, and (ii) failed experiences that summarize failure modes and actionable avoidance rules. Each experience is stored in a structured format (summary, suggestions, applicable conditions), and is additionally indexed by a semantic

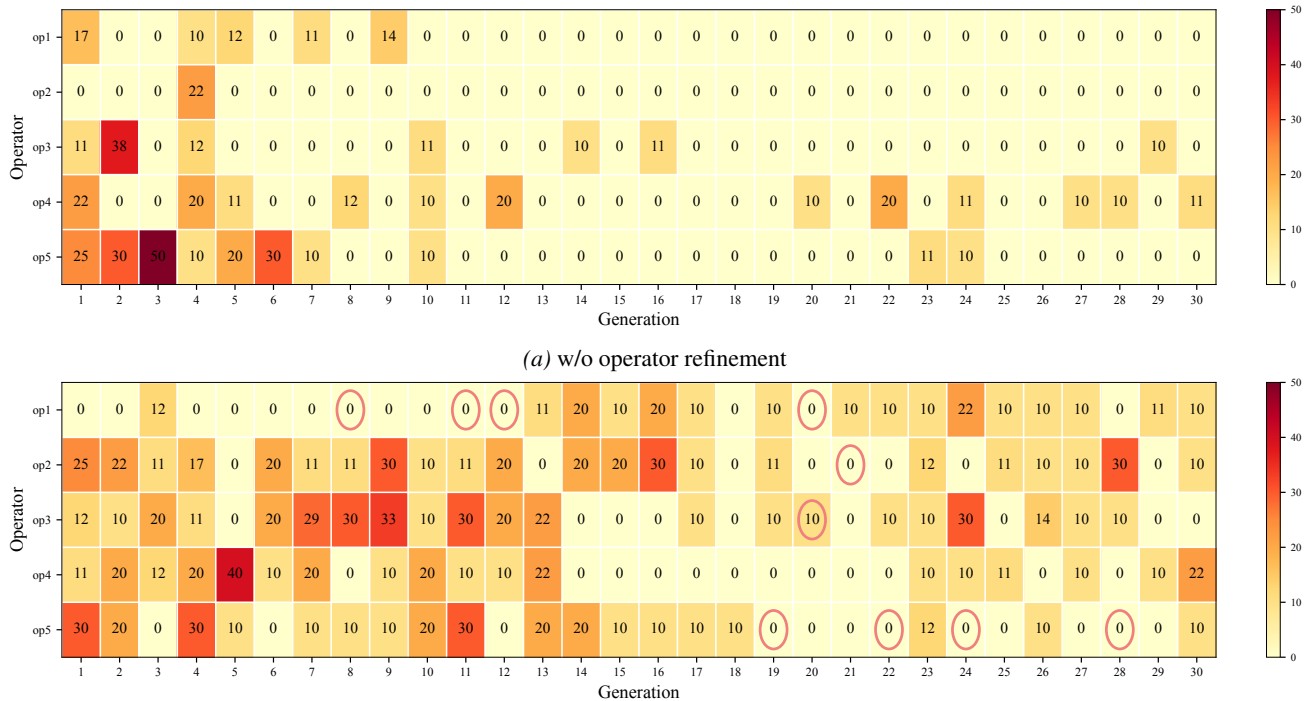

*(a)* w/o operator refinement

*(b)* with operator refinement only. Red circles indicate radical refinements.

*Figure 5.* Operator survival heatmaps. Operator refinement prevents operator-level stagnation by reviving temporarily underperforming operators.

embedding for efficient similarity search.

**Trajectory and Experiences.** After applying an operator to parent individuals and evaluating the produced offspring, we form a *trajectory* record that captures the essential context needed for reflection: the operator goal, parents/offspring descriptions, the objective type, and the measured improvement with respect to the best parent. This trajectory is the sole input to the Reflector, ensuring that reflections are grounded in observed outcomes rather than speculative reasoning. If the offspring yields a meaningful improvement, the Reflector generates a small set of successful experience entries that distill what changed and why it helped; otherwise, it generates failed experience entries that diagnose what likely went wrong and how to avoid repeating it. We store these entries separately so that retrieval can explicitly balance "what to do" and "what not to do".

**Example for Trajectory**

```
{
    "operator_goal": "create a new algorithm that has a totally different form
    from the given ones but can be motivated from them",

    "parents": [{"objective": 9.88497, "algorithm": "Combines dynamic
    look-ahead depth with adaptive weight adjustment based on remaining nodes
    count, using exponential decay for look-ahead importance and linear
    scaling for immediate distance priority."},
    {"objective": 10.28278, "algorithm": "This algorithm combines greedy
    nearest-neighbor selection with a two-step lookahead mechanism that
    evaluates potential future moves by considering both immediate distance
    and the optimal subsequent step from each candidate node."}],
```

```
      "offspring": {"objective": 8.64145, "algorithm": "Combines dynamic
      candidate filtering with lookahead optimization, first selecting
      promising candidates via distance thresholding then performing
      depth-limited optimal path evaluation from each candidate."},
      "objective_type": "min", "best_parent_objective": 9.88497,
      "improvement": 1.2435199999999984, "problem_name": "tsp_constructive"
}
```

## Example for Structured Experience

```
{
    "is_success": true,

    "distilled_items": "[{"summary": "Dynamic candidate filtering before
    lookahead improves efficiency", "recommendations": ["First filter
    candidates based on distance thresholds to reduce search space",
    "Apply depth-limited optimal path evaluation only to promising candidates",
    "Adjust filtering thresholds dynamically based on problem size"],
    "applicable_when": "When dealing with large search spaces requiring
    lookahead evaluation"},
    {"summary": "Combining immediate distance with lookahead yields better
    solutions", "recommendations": ["Use distance thresholding for initial
    candidate selection", "Apply limited-depth lookahead only to top
    candidates","Balance computational budget between filtering and evaluation
    stages"], "applicable_when": "When needing to balance solution quality and
    computational cost"},
    {"summary": "Multi-stage evaluation outperforms single-metric approaches",
    "recommendations": ["Implement staged evaluation with increasing
    computational intensity", "Use coarse metrics first to filter,
    then detailed evaluation", "Dynamically adjust evaluation depth based on
    remaining nodes\"], "applicable_when": "When computational resources
    need to be strategically allocated"}]"

    "utility_score": 2
}
```

**Query and Retrieval.** Before generating a new offspring, the Evolver constructs a retrieval query from the current operator goal and the parents context. We then retrieve top-$k$ items from both banks by semantic similarity using cosine similarity and inject them into the evolution prompt. In all experiments, we set $k = 3$. Specifically, given two experience vectors $e_1$ and $e_2$, their similarity is computed as:

$$\text{sim}(e_1, e_2) = \frac{\mathbf{e_1} \cdot \mathbf{e_2}}{\|\mathbf{e_1}\|\|\mathbf{e_2}\|}$$

This bidirectional retrieval provides two benefits: it encourages reuse of effective motifs that have worked under similar operator goals, while simultaneously constraining the search away from known degenerate edits or brittle heuristics that previously failed.

> **Example for Query**
>
> Operator goal: create a new algorithm that has a totally different form from the given ones but can be motivated from them. Parent 1 (objective=9.88497): Combines dynamic look-ahead depth with adaptive weight adjustment based on remaining nodes count, using exponential decay for look-ahead importance and linear scaling for immediate distance priority. Parent 2 (objective=10.28278): This algorithm combines greedy nearest-neighbor selection with a two-step lookahead mechanism that evaluates potential future moves by considering both immediate distance and the optimal subsequent step from each candidate node.

**Dynamic Maintenance via Credit Assignment.** To keep the pool reliable as it grows, we adopt a simple fully specified utility-based management rule. Each experience entry $e$ is initialized with a utility score $u(e) = 1$. When $e$ is retrieved and included in the prompt for generating a candidate offspring $x$, we compare $x$ with its best parent $x_{\text{parent}}$ in that iteration using the task objective $f(\cdot)$. We mark the retrieval as successful if $f(x)$ strictly improves over $f(x_{\text{parent}})$, denoting lower values for minimization tasks and higher for maximization tasks. Then we update $u(e) \leftarrow u(e) + 1$ if successful and $u(e) \leftarrow u(e) - 1$ otherwise. If $u(e) < 0$, we delete $e$ from the pool.

When multiple experiences are retrieved for one generation, we apply the same success or failure signal to each retrieved entry.

Additionally, to ensure the quality of the experience pool, we use cosine similarity to evaluate the relevance of retrieved experiences by comparing the current query embedding with those of stored entries. If a newly retrieved candidate experience contradicts an already stored experience under the same (or highly similar) context, we discard the new candidate (i.e., do not add it to the pool and do not use it for subsequent retrieval). Meanwhile, we penalize the existing stored entry by decreasing its utility score by 1 $(u(e_{old}) \leftarrow u(e_{old}) - 1)$, reflecting that it may be misleading in this context.

## F. Statistical Analysis

To assess the stability of RefineEvo, we report the mean and standard deviation over 3 independent runs for the main benchmarks. Tables 15 and 16 compare RefineEvo against PartEvo and OpenEvolve.

*Table 15.* TSP constructive heuristics: mean $\pm$ std over 3 independent runs.

| Method | $n$=100 | $n$=200 | $n$=500 | TSPLIB (gap%) |
|---|---|---|---|---|
| PartEvo | $8.701 \pm 0.093$ | $12.137 \pm 0.116$ | $18.854 \pm 0.180$ | $13.44 \pm 0.128$ |
| OpenEvolve | $8.639 \pm 0.042$ | $12.074 \pm 0.053$ | $18.792 \pm 0.067$ | $12.92 \pm 0.046$ |
| **RefineEvo** | $\mathbf{8.541 \pm 0.031}$ | $\mathbf{11.868 \pm 0.034}$ | $\mathbf{18.476 \pm 0.049}$ | $\mathbf{11.54 \pm 0.023}$ |

*Table 16.* Online BPP: mean excess-bin ratio $\pm$ std over 3 independent runs.

| Method | 5k_C100 | 10k_C100 | 100k_C100 |
|---|---|---|---|
| PartEvo | $0.34\% \pm 0.05\%$ | $0.22\% \pm 0.02\%$ | $0.02\% \pm 0.01\%$ |
| OpenEvolve | $0.28\% \pm 0.03\%$ | $\mathbf{0.16\% \pm 0.01\%}$ | $0.01\% \pm 0.01\%$ |
| **RefineEvo** | $\mathbf{0.25\% \pm 0.03\%}$ | $0.19\% \pm 0.02\%$ | $\mathbf{0.01\% \pm 0.01\%}$ |

## G. Detailed Ablation Results on Hyperparameter $k$

In this section, we provide the full sensitivity analysis for the hyperparameter $k$ regarding Operator Selection (budget size) and Operator Refinement (update frequency). Table 17 details the performance across all three benchmarks.

**Operator Selection with varying $k$.** We compare Random Selection against Planner Selection by varying the budget $k \in \{1, 2, 3, 4\}$.

- **Impact on TSPLIB & TSP100:** Increasing the budget generally improves solution quality for both methods. However, the Planner consistently outperforms Random Selection at comparable budget levels. At the optimal setting of $k = 4$, Planner Selection achieves 13.15% on TSPLIB and 8.603 on TSP100, significantly surpassing the best Random Selection results (15.14% and 8.639, respectively).

- **Impact on BPP:** The Planner demonstrates superior generalization on the BPP task as well. At $k = 4$, Planner Selection achieves 0.27%, outperforming the best Random Selection result of 0.30%. This consistent superiority across diverse tasks confirms that the Planner identifies effective operator subsets that generalize well, rather than overfitting to specific problem instances.

**Operator Refinement with varying interval $k$.** We test a Fixed-Interval Refinement strategy where operators are blindly refined every $k$ generations.

- **Instability across Tasks:** The method shows high sensitivity and instability. For instance, on BPP, setting $k = 4$ causes a drastic performance spike to 1.71%, which is far worse than the 0.25% of RefineEvo. Similarly, on TSPLIB, a frequent update of $k = 2$ leads to a poor gap of 22.41%.

- **Overall Inferiority:** Even at the best fixed setting ($k = 3$), the approach yields 14.05% on TSPLIB and 8.651 on TSP100, lagging behind RefineEvo (11.54% and 8.541). This demonstrates that a rigid schedule cannot adapt to the varying difficulties of different tasks, whereas the event-triggered mechanism of RefineEvo maintains consistent superiority.

*Table 17.* Full ablation results for varying $k$. Random Selection and Planner Selection choose exactly $k$ operators per generation. Fixed-Interval Refinement triggers updates every $k$ generations. **Bold** indicates the best result within each category.

| Category | Setting | TSPLIB ↓ | TSP100 ↓ | BPP 5k_C100 ↓ |
|---|---|---|---|---|
| RefineEvo (Dynamic) | - | 11.54% | 8.541 | 0.25% |
| Random Selection | $k = 1$ | 18.08% | 8.948 | 0.52% |
| | $k = 2$ | 17.73% | 8.836 | 0.43% |
| | $k = 3$ | 17.70% | 8.787 | 0.40% |
| | $k = 4$ (Best) | **15.14%** | **8.639** | **0.30%** |
| Planner Selection | $k = 1$ | 17.16% | 8.907 | 0.43% |
| | $k = 2$ | 15.94% | 8.673 | 0.40% |
| | $k = 3$ | 14.95% | 8.657 | 0.40% |
| | $k = 4$ (Best) | **13.15%** | **8.603** | **0.27%** |
| Random Mode | - | 14.74% | 8.665 | 0.27% |
| Fixed-Interval Refinement | $k = 2$ | 22.41% | 8.945 | 0.51% |
| | $k = 3$ (Best) | **14.05%** | **8.651** | **0.28%** |
| | $k = 4$ | 16.64% | 8.668 | 1.71% |
| | $k = 5$ | 14.73% | 8.727 | 0.42% |

### G.1. RefineEvo with Other LLMs

To examine whether the advantages of RefineEvo depend on a specific LLM backbone, we rerun the full pipeline by replacing the default LLM with `gpt-4o-mini`. We apply the same replacement to all LLM-based AHD baselines to ensure a fair comparison. All other settings (prompts, evaluation budget, termination criteria, and decoding hyperparameters) follow the main experimental protocol. Table 18 reports the results.

*Table 18.* RefineEvo and baselines with `gpt-4o-mini` as the LLM backbone. Boldface indicates the best result.

| Methods | TSP (Obj. ↓) | | | KP (Obj. ↑) | | | BPP (Gap. ↓) | | |
|---------|-------|-------|-------|-------|-------|-------|-------|-------|-------|
| | $n=100$ | $n=200$ | $n=500$ | $n=100$ | $n=200$ | $n=500$ | $5k\_C100$ | $10k\_C100$ | $100k\_C100$ |
| FunSearch | 8.845 | 12.310 | 19.342 | 40.632 | 57.811 | 90.631 | 1.89% | 1.79% | 1.64% |
| EoH | 8.841 | 12.308 | 19.317 | 40.655 | 57.833 | 90.683 | 0.63% | 0.31% | **0.01%** |
| ReEvo | 8.821 | 12.349 | 19.189 | 40.638 | 57.827 | 90.672 | 0.73% | 0.41% | 0.15% |
| HSEvo | 8.735 | 12.209 | 19.269 | 40.669 | 57.838 | 90.631 | 0.71% | 0.25% | 0.04% |
| MCTS-AHD | 8.773 | 12.253 | 19.103 | 40.662 | 57.843 | 90.686 | **0.36%** | 0.20% | 0.02% |
| **RefineEvo** | **8.712** | **12.178** | **19.101** | **40.676** | **57.849** | **90.687** | **0.36%** | **0.19%** | **0.01%** |

Overall, RefineEvo remains consistently strong under a smaller-capacity backbone. It achieves the best objective values on all TSP scales, and obtains the best performance on BPP gaps across all instance regimes. On KP, where the absolute differences among methods are small under this backbone, RefineEvo still attains the top scores across all three scales.

We further evaluate with GPT-4o as the backbone. Table 19 reports the results. RefineEvo achieves the best objective values on all TSP scales and the lowest BPP ratios, consistent with the trends observed under DeepSeek-V3 and GPT-4o-mini.

*Table 19.* RefineEvo and baselines with GPT-4o as the LLM backbone. **Bold** indicates the best result.

| Methods | TSP (Obj. ↓) | | | BPP (Ratio ↓) | | |
|---------|-------|-------|-------|-------|-------|-------|
| | $n=100$ | $n=200$ | $n=500$ | 5k\_C100 | 10k\_C100 | 100k\_C100 |
| EoH | 8.724 | 12.194 | 19.129 | 0.48% | 0.39% | 0.25% |
| ReEvo | 8.569 | 11.929 | 18.584 | 0.71% | 0.25% | 0.04% |
| **RefineEvo** | **8.534** | **11.919** | **18.503** | **0.43%** | **0.22%** | **0.01%** |

## G.2. Scaling to Stronger Backbones

To investigate how backbone capability affects evolutionary search, we additionally evaluate RefineEvo with Claude Opus 4.6, a substantially stronger model. Table 20 compares the improvement from initialization to the final evolved heuristic on the TSP100 training set across four backbones. The evolutionary process improves the best individual by 7.77%–11.88% across all backbones, indicating that evolutionary search remains beneficial even as backbone capability increases. Notably, Claude Opus 4.6 achieves the largest improvement (11.88%), suggesting that stronger models can benefit even more from the structured search process.

*Table 20.* Evolution improvement on the TSP100 training set across LLM backbones.

| Backbone | Init best | Evolved best | Init mean | Evolved mean |
|----------|-----------|--------------|-----------|--------------|
| GPT-4o-mini | 9.370 | 8.643 (↓7.77%) | 10.426 | 8.675 (↓16.79%) |
| DeepSeek-V3 | 9.317 | 8.536 (↓8.38%) | 10.803 | 8.572 (↓20.65%) |
| GPT-4o | 9.283 | 8.518 (↓8.24%) | 10.609 | 8.595 (↓18.99%) |
| Claude Opus 4.6 | 9.106 | 8.024 (↓11.88%) | 10.137 | 8.036 (↓20.72%) |

Table 21 reports cross-scale generalization. Stronger backbones can further improve the evolved heuristics in several settings, with Claude Opus 4.6 achieving a TSPLIB gap of only 5.45%. Importantly, even GPT-4o-mini after evolution (TSPLIB: 16.57%) substantially outperforms Claude Opus 4.6 without evolution (17.75%), suggesting that evolutionary search and backbone capability are complementary rather than substitutional.

We note that Claude Opus 4.6 (USD 25/M output tokens) costs approximately 22.7× more than DeepSeek-V3 (USD 1.10/M output tokens). Since evolutionary search involves repeated LLM calls over many generations, this pricing gap is practically significant. DeepSeek-V3 with evolution (TSPLIB: 11.54%) substantially outperforms Claude Opus 4.6 without evolution (TSPLIB: 17.75%), making cost-efficient backbones with evolutionary search a practical and effective choice.

*Table 21.* Cross-scale generalization across LLM backbones on constructive TSP.

| Backbone | TSP100 ↓ | TSP200 ↓ | TSP500 ↓ | TSPLIB (gap%) ↓ |
|---|---|---|---|---|
| GPT-4o-mini (evolved) | 8.712 | 12.178 | 19.101 | 16.57 |
| DeepSeek-V3 (evolved) | 8.541 | 11.868 | 18.476 | 11.54 |
| GPT-4o (evolved) | 8.534 | 11.919 | 18.503 | – |
| Claude Opus 4.6 (evolved) | **8.018** | **11.202** | **17.478** | **5.45** |
| Claude Opus 4.6 (init best, no evolution) | 9.321 | 13.013 | 19.828 | 17.75 |

## G.3. The Universality of Experience

This section investigates the *cross-problem universality* of the Bidirectional Experience Pool (BEP) under our constructive heuristics framework. Intuitively, many experiences in RefineEvo are not tied to a single task instance distribution, but rather describe *general heuristic transformation principles* (e.g., how to incorporate greedy signals, how to repair infeasible partial solutions, how to tune randomness vs. determinism, or how to avoid a known failure mode under certain population states). If such principles are captured in a structured way, they may transfer across problem types despite differences in objective definition.

**Warm-start protocol.** We consider three problems: TSP, KP, and Online BPP. Different from the default *cold-start* setting where BEP is empty at the beginning of evolution, here we *initialize* (warm-start) the experience pool using experiences collected from a *single other* problem, and then run RefineEvo on the target problem. Importantly, we do not mix sources: for each target task, we perform independent evolutionary runs, each warm-started by experiences from one of the other two tasks; we also include the cold-start baseline.

To isolate the effect of experience transfer, all other components of RefineEvo are kept identical to the default setting. During warm-start, we directly preload the BEP with the stored experience items (including both positive and negative entries) from the *source* task; at the start of the target-task evolution, RefineEvo can retrieve these items based on the query constructed from the current state and operator intent, and it can also continue to collect new experiences online.

**Evaluation.** We evaluate the final best heuristic from each run on two test scales ("Small" and "Large") for each task to assess whether transferred experience helps generalization across sizes. For TSP and KP, we use $n = \{100, 500\}$; for Online BPP, we use #items $= \{5k, 10k\}$. Table 7 summarizes the cross-task transfer results.

**Results.** We highlight three takeaways. (1) Transfer is partially universal but task compatibility matters. Warm-starting with experiences from a *related* task can be beneficial or neutral, whereas transferring from a less compatible task may hurt. This aligns with the fact that some experiences encode problem-specific structural assumptions. (2) The gains are more visible on the larger scale. When transfer helps, it typically improves (or preserves) performance more on the "Large" regime, suggesting that reusable experiences can provide robust guidance that generalizes across instance sizes. (3) BEP is resilient to imperfect transfer. Even when warm-start experiences are mismatched, RefineEvo does not catastrophically fail; the Planner and subsequent online experience accumulation allow the search to gradually override unhelpful prior items.

Overall, these results indicate that the experience pool contains both *task-agnostic* heuristic knowledge and *task-specific* priors. This motivates future work on automatically estimating experience-task compatibility to maximize safe transfer.

## G.4. BEP Overhead Analysis

We analyze the computational overhead of the Bidirectional Experience Pool. Table 24 reports the per-iteration time breakdown on the TSP constructive benchmark. Semantic retrieval (query embedding plus cosine similarity) accounts for only 0.36 s (1.55% of total time). The experience-related cost is dominated by the Reflector's LLM call for extracting structured insights (9.03 s, 38.77%). As shown in Table 6, removing experiences degrades TSPLIB performance from 11.54% to 15.91%, indicating that the additional cost is compensated by the performance gain.

## G.5. Token Cost Breakdown

Table 22 provides detailed token statistics for each method on the TSP constructive benchmark under a budget of 1000 evaluations. RefineEvo's total cost ($\sim$511K tokens) is among the lowest, comparable to HSEvo ($\sim$497K) and PartEvo ($\sim$507K) while achieving stronger final performance in the main experiments. The additional overhead from the Planner and Reflector is offset by selective operator scheduling, which avoids redundant LLM calls on inactive operators.

*Table 22.* Token usage comparison across LLM-based AHD methods on TSP constructive heuristics.

| Method | Total completion tokens |
|---|---|
| HSEvo | $\sim$497K |
| PartEvo | $\sim$507K |
| RefineEvo | $\sim$511K |
| MCTS-AHD | $\sim$585K |
| EoH | $\sim$595K |
| ReEvo | $\sim$751K |

## G.6. CVRPLib Generalization

To evaluate out-of-distribution generalization on vehicle routing, we test the evolved heuristics on CVRPLib benchmark instances under both constructive and ACO paradigms. Table 23 reports the average optimality gap for each method. RefineEvo achieves the best generalization under both paradigms. The advantage is particularly pronounced under ACO (16.91% vs. OpenEvolve's 21.13%), where the evolved heuristic information component can better exploit the richer search dynamics of the metaheuristic framework.

*Table 23.* Generalization on CVRPLib benchmark instances. The metric is the average optimality gap (%, lower is better).

| Method | Constructive (gap%) | ACO (gap%) |
|---|---|---|
| EoH | 40.08 | 33.33 |
| ReEvo | 35.72 | 31.91 |
| HSEvo | – | 26.62 |
| MCTS-AHD | 40.71 | 34.36 |
| PartEvo | 34.70 | – |
| OpenEvolve | 32.62 | 21.13 |
| **RefineEvo** | **29.34** | **16.91** |

*Table 24.* Per-iteration time breakdown of RefineEvo on TSP constructive heuristics.

| Component | Time (s) | Proportion |
|---|---|---|
| Operator selection (Planner) | 2.41 | 10.35% |
| Heuristic generation (LLM) | 11.49 | 49.33% |
| Semantic retrieval | 0.36 | 1.55% |
| Reflection (LLM) | 9.03 | 38.77% |

## G.7. Detailed Results on TSPLIB

We further evaluate the step-by-step constructive heuristics on the real-world TSPLIB benchmark. Following standard practice, we report the *optimality gap* in percentage on each instance:

$$\text{Gap}(\%) \;=\; 100 \times \frac{f(h) - f^\star}{\max(|f^\star|, \epsilon)},$$

where $f(h)$ is the tour length produced by heuristic $h$, $f^\star$ is the best-known tour length reported in TSPLIB, and $\epsilon$ avoids numerical issues. Lower is better.

We include Greedy construction, LLM-based AHD baselines and **RefineEvo**, we evaluate the final discovered heuristic under the same TSPLIB protocol. TSPLIB constructive heuristics can depend on the starting node. To reduce this sensitivity,

for each method and each TSPLIB instance we run the heuristic multiple times with different starting nodes and report the average gap. Unless otherwise stated, the reported value is averaged over three runs with distinct starting nodes. For each AHD method, we select its final heuristic as the best-performing one among multiple independent search runs, consistent with the main paper.

Table 25 and Figure 6 reports per-instance gaps. Overall, RefineEvo achieves the lowest average gap and wins on the largest number of instances among LLM-designed heuristics, while also being competitive with classical constructive heuristics and the AHD baseline. We additionally observe that cross-instance robustness improves when the discovered heuristic maintains stable behavior across starting nodes, which aligns with our design goals of experience-guided evolution.

*Table 25.* TSPLIB results (optimality gap %) compared to LLM-based AHD and classical heuristics.

| Instance | Greedy | POMO | FunSearch | EoH | ReEvo | HSEvo | MCTS-AHD | RefineEvo |
|---|---|---|---|---|---|---|---|---|
| ts225 | 16.80 | 8.58 | 6.03 | **4.11** | 6.56 | 6.03 | 9.09 | 9.65 |
| rat99 | 21.80 | **4.86** | 18.91 | 13.52 | 12.41 | 12.67 | 12.39 | 9.83 |
| rl1889 | 23.70 | 88.50 | 28.24 | 26.00 | 17.50 | 18.22 | 15.50 | **14.00** |
| u1817 | 22.20 | 84.91 | 22.63 | 18.27 | 16.64 | 22.63 | 17.16 | **11.60** |
| d1655 | 23.90 | 93.10 | 38.34 | 22.83 | 17.51 | 18.34 | 17.35 | **10.90** |
| bier127 | 23.30 | 19.88 | 22.11 | 21.06 | 10.79 | 12.11 | 12.06 | **7.76** |
| lin318 | 25.80 | **11.26** | 22.83 | 21.42 | 16.63 | 12.83 | 15.19 | 15.05 |
| eil51 | 32.00 | **0.83** | 8.90 | 7.98 | 6.47 | 8.90 | 4.75 | 9.72 |
| d493 | 24.00 | 67.79 | 22.36 | 22.65 | 13.43 | 12.36 | 12.56 | **10.06** |
| kroB100 | 26.30 | **0.48** | 14.89 | 12.78 | 12.20 | 14.89 | 13.84 | 13.61 |
| kroC100 | 25.80 | **1.14** | 16.13 | 13.22 | 15.88 | 16.13 | 16.84 | 8.17 |
| ch130 | 25.70 | **0.51** | 10.03 | 6.21 | 9.40 | 10.03 | 11.26 | 10.04 |
| pr299 | 31.40 | 16.47 | 28.26 | 27.80 | 20.63 | 17.26 | 16.07 | **10.26** |
| fl417 | 32.40 | 22.93 | 25.80 | 38.83 | 19.15 | 15.80 | 16.35 | **15.45** |
| d657 | 29.70 | 60.14 | 36.52 | 32.78 | 16.04 | 16.52 | 16.06 | **13.24** |
| kroA150 | 26.10 | **2.27** | 14.50 | 13.87 | 11.62 | 14.50 | 12.99 | 12.04 |
| fl1577 | 25.00 | 88.16 | 22.39 | 16.98 | **12.11** | 12.39 | 16.43 | 14.48 |
| u724 | 28.50 | 36.58 | 26.35 | 21.01 | 16.87 | 16.35 | 16.73 | **13.09** |
| pr264 | 17.90 | 18.65 | 15.02 | 15.67 | 16.78 | 15.02 | 16.53 | **13.14** |
| pr226 | 24.60 | 5.17 | 21.37 | 26.93 | 18.02 | 21.37 | 16.63 | **4.06** |
| pr439 | 27.40 | 27.56 | 20.26 | 17.70 | 19.25 | 20.26 | 19.30 | **16.29** |
| Average Gap | 25.443 | 31.418 | 21.041 | 19.125 | 14.566 | 14.981 | 14.528 | **11.545** |

## G.8. Detailed Results of ACO

**Heuristic measure in ACO.** Ant Colony Optimization (ACO) alternates between *stochastic solution construction* and *pheromone trail updates*. In each construction step, an ant located at node $i$ selects the next node $j$ from a feasible candidate set $\mathcal{N}(i)$ according to a transition probability that combines pheromone intensity $\tau_{ij}$ and a heuristic desirability (a.k.a. heuristic measure) $\eta_{ij}$:

$$p_{ij} = \frac{\tau_{ij}^{\alpha} \eta_{ij}^{\beta}}{\sum\limits_{k \in \mathcal{N}(i)} \tau_{ik}^{\alpha} \eta_{ik}^{\beta}}, \tag{15}$$

where $\alpha$ and $\beta$ control the relative importance of pheromone and heuristic guidance. In our setting, the learnable component is the heuristic measure $\eta_{ij}$, produced by the discovered heuristic function. To ensure numerical stability, we clamp $\eta_{ij}$ to be positive, e.g., $\eta_{ij} \leftarrow \max(\eta_{ij}, \epsilon)$, and keep all other ACO components fixed across methods.

**Pheromone update.** After all ants construct solutions in an iteration, we update pheromones using a standard evaporation-and-deposit rule:

$$\tau_{ij} \leftarrow (1 - \rho)\tau_{ij} + \Delta\tau_{ij}, \qquad \Delta\tau_{ij} = \sum_{a \in \mathcal{A}} \Delta\tau_{ij}^{(a)}, \tag{16}$$

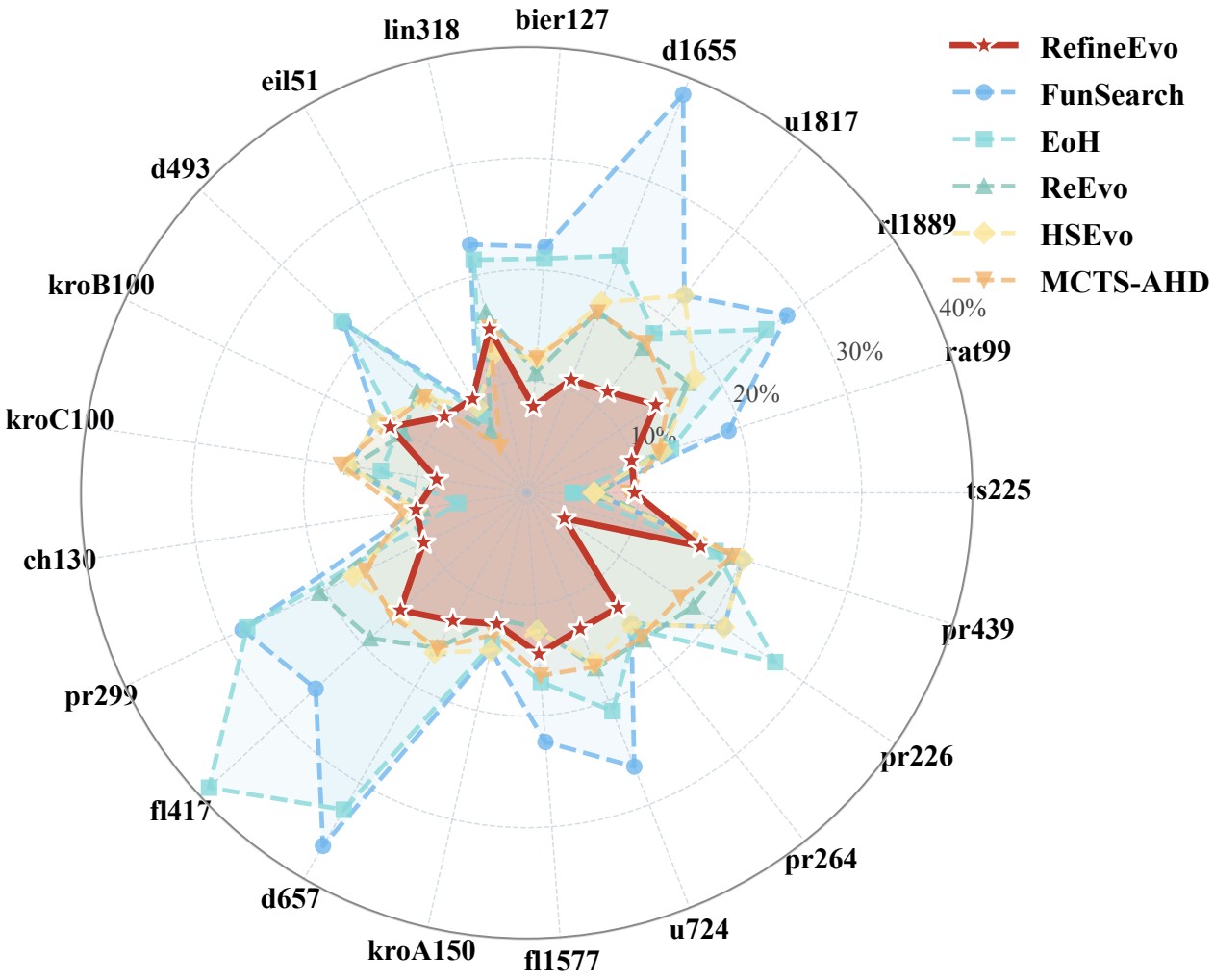

*Figure 6.* TSPLIB instance gap.

where $\rho$ is the evaporation rate and $\Delta\tau_{ij}^{(a)}$ depends on the solution quality of ant $a$. We use the same update variant, initialization, candidate set policy, and feasibility handling for all compared methods; hence, performance differences primarily reflect the quality of the learned $\eta_{ij}$.

*Table 26.* ACO hyperparameters for heuristic evaluation and test-time convergence plots.

| Problem | #Ants | Iter. (evolution) | Iter. (test) |
|---|---|---|---|
| TSP/CVRP | 30 | 100 | 200 |
| MKP | 10 | 50 | 100 |

**Convergence behavior.** Figure 7 reports the convergence trajectories of ACO under different heuristic measures. For each method, we plot the *best-so-far* objective as a function of ACO iterations, averaged over the test instances and random seeds. Compared to baselines, the heuristic discovered by RefineEvo typically improves both (i) *early-stage progress* (faster reduction in objective) and (ii) *final convergence quality* under a sufficiently large iteration budget, indicating that the learned $\eta_{ij}$ provides consistently informative guidance throughout the sampling-and-update process rather than only yielding short-horizon gains.

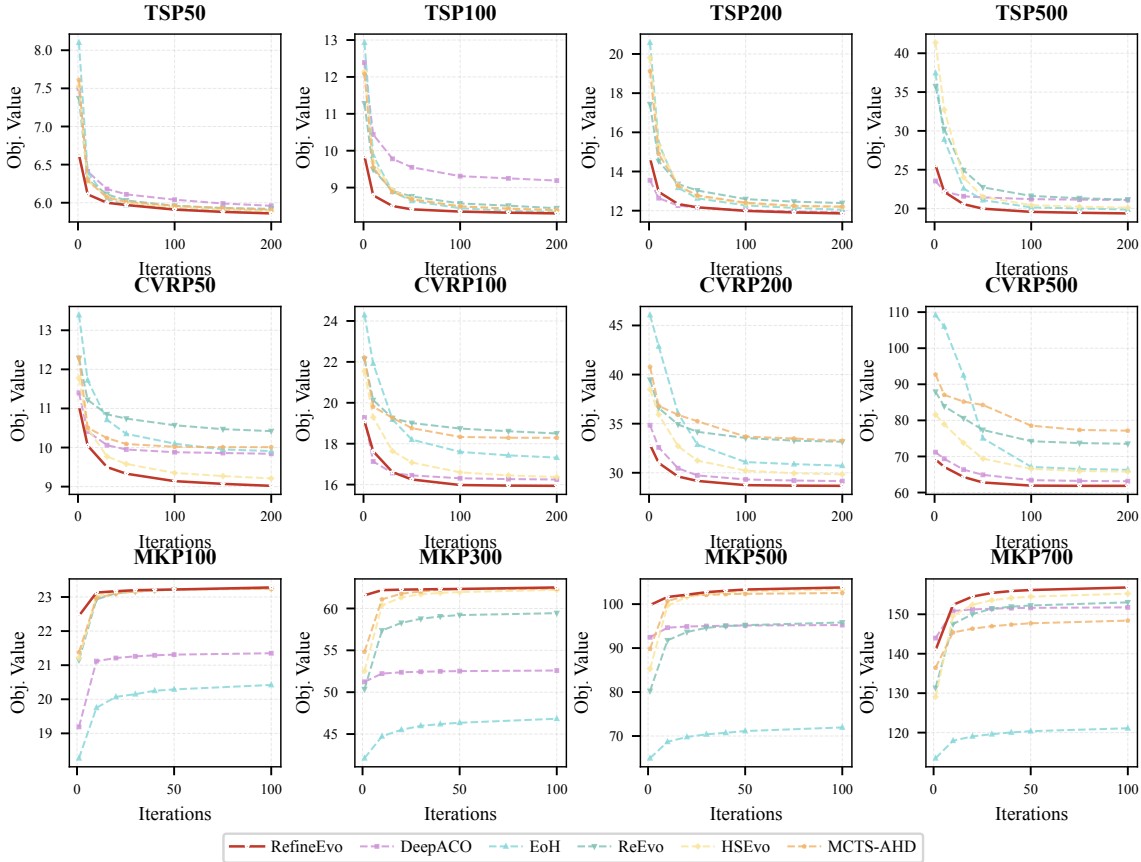

*Figure 7.* ACO convergence curves under different heuristic measures.

## G.9. Generated heuristics

### G.9.1. COMPARISON

**TSP Constructive Heuristic (RefineEvo)**

**Description** Hybrid algorithm combining adaptive lookahead with destination-aware dynamic candidate filtering, phased regret-progressive evaluation, and strategic horizon variation while maintaining computational efficiency through intelligent threshold adjustment and adaptive scoring weights.

**Code**

```python
import numpy as np
from collections import defaultdict

def select_next_node(current_node, destination_node, unvisited_nodes,
    distance_matrix):
    remaining_count = len(unvisited_nodes)
    if remaining_count == 1:
        return unvisited_nodes.pop()

    # Enhanced dynamic threshold with strategic destination-awareness
    current_distances = distance_matrix[current_node]
```

```python
    min_dist = min(current_distances[n] for n in unvisited_nodes)
    progress = 1 - (remaining_count / (distance_matrix.shape[0] - 1))
    dest_ratio = distance_matrix[current_node][destination_node] /
    np.median(current_distances)
    dynamic_threshold = min_dist * (1.4 + 0.5 * progress) * (1 + 0.3 * dest_ratio)

    # Intelligent candidate filtering with progressive regret evaluation
    candidates = set()
    regret_scores = {}
    for node in unvisited_nodes:
        dist = current_distances[node]
        if dist <= dynamic_threshold:
            candidates.add(node)
        else:
            sorted_remain = sorted(unvisited_nodes - {node}, key=lambda x:
    current_distances[x])
            if len(sorted_remain) > 1:
                regret = current_distances[sorted_remain[1]] - dist
                if regret > 0.25 * dynamic_threshold * (0.9 + 0.3 * progress):
                    candidates.add(node)
                    regret_scores[node] = regret

    if not candidates:
        candidates = unvisited_nodes

    # Strategic horizon variation with computational efficiency
    max_depth = min(4, max(2, int(2.5 + progress * 1.5)))
    evaluation_cache = defaultdict(dict)
    best_node = None
    best_score = float('inf')

    for candidate in candidates:
        remaining = unvisited_nodes - {candidate}
        path_distance = current_distances[candidate]
        regret_sum = regret_scores.get(candidate, 0)
        depth = 0
        current_path = [current_node, candidate]

        while depth < max_depth and remaining:
            cache_key = (current_path[-1], frozenset(remaining))
            if depth in evaluation_cache and cache_key in evaluation_cache[depth]:
                next_node, dist, regret = evaluation_cache[depth][cache_key]
            else:
                sorted_remain = sorted(remaining, key=lambda x:
    distance_matrix[current_path[-1]][x])
                next_node = sorted_remain[0]
                dist = distance_matrix[current_path[-1]][next_node]
                regret = (distance_matrix[current_path[-1]][sorted_remain[1]] -
    dist) if len(sorted_remain) > 1 else 0
                evaluation_cache[depth][cache_key] = (next_node, dist, regret)

            path_distance += dist
            regret_sum += regret * (1.1 - depth/max_depth)  # Progressive decay
            current_path.append(next_node)
            remaining.remove(next_node)
            depth += 1

        # Efficient completion with destination consideration
        last_node = current_path[-1]
        while remaining:
            nearest = min(remaining, key=lambda x: distance_matrix[last_node][x])
```

```
        path_distance += distance_matrix[last_node][nearest]
        last_node = nearest
        remaining.remove(nearest)

    if last_node != destination_node:
        path_distance += distance_matrix[last_node][destination_node] * (0.5 +
0.5 * progress)

    # Adaptive scoring with strategic weights
    lookahead_weight = 0.55 + 0.15 * min(max_depth, 3)
    score = (0.45 * current_distances[candidate] +
            lookahead_weight * path_distance +
            (0.55 - lookahead_weight) * regret_sum)

    if score < best_score:
        best_score = score
        best_node = candidate

return best_node if best_node is not None else min(unvisited_nodes, key=lambda
x: current_distances[x])
```

## TSP Constructive Heuristic (EoH)

**Code**

```python
import math

def select_next_node(current_node, destination_node, unvisited_nodes,
    distance_matrix):
    next_node = None
    best_score = -float('inf')
    for node in unvisited_nodes:
        immediate_dist = distance_matrix[current_node][node]
        future_dist = distance_matrix[node][destination_node]
        look_ahead_dists = sorted([distance_matrix[node][u] for u in
    unvisited_nodes if u != node])
        look_ahead_sin = sum(d * math.sin(i + 1) / (i + 1) for i, d in
    enumerate(look_ahead_dists[:5]))
        look_ahead_log = sum(d / math.log(i + 2) for i, d in
    enumerate(look_ahead_dists[5:10]))
        score = (0.35 * future_dist + 0.4 * look_ahead_sin + 0.25 *
    look_ahead_log) / (immediate_dist ** 0.6)
        if score > best_score:
            best_score = score
            next_node = node
    return next_node
```

## TSP Constructive Heuristic (ReEvo)

**Code**

```python
import numpy as np
import numpy as np
from collections import defaultdict
```

```python
def select_next_node(current_node, destination_node, unvisited_nodes,
    distance_matrix):
    if not unvisited_nodes:
        return destination_node

    candidates = np.array(list(unvisited_nodes))
    n_remaining = len(candidates)

    # Phase 1: Fast nearest neighbor filter
    current_dists = distance_matrix[current_node, candidates]
    nn_candidate = candidates[np.argmin(current_dists)]
    if n_remaining <= 5:  # Small problems use pure NN
        return nn_candidate

    # Dynamic parameters
    candidate_pool = min(15, max(5, int(np.sqrt(n_remaining))))
    lookahead_base = min(2, max(1, int(np.log(n_remaining)/2)))

    # Cached MST approximations
    mst_cache = defaultdict(float)

    def get_mst(nodes):
        key = frozenset(nodes)
        if key not in mst_cache:
            if len(nodes) <= 1:
                mst_cache[key] = 0
            else:
                submatrix = distance_matrix[np.ix_(nodes, nodes)]
                sorted_dists = np.sort(submatrix[np.triu_indices(len(nodes), 1)])
                mst_cache[key] = np.sum(sorted_dists[:len(nodes)-1])
        return mst_cache[key]

    # Phase 2: Hybrid evaluation
    sorted_idx = np.argsort(current_dists)
    top_candidates = candidates[sorted_idx[:candidate_pool]]

    best_node = nn_candidate
    min_total = current_dists[sorted_idx[0]] + distance_matrix[nn_candidate,
    destination_node] if n_remaining == 1 else np.inf

    for i, node in enumerate(top_candidates):
        step_cost = current_dists[sorted_idx[i]]
        if step_cost >= min_total:
            continue

        remaining = unvisited_nodes - {node}
        if not remaining:
            total_cost = step_cost + distance_matrix[node, destination_node]
            if total_cost < min_total:
                min_total = total_cost
                best_node = node
            continue

        # Adaptive hybrid estimation
        remaining_list = list(remaining)
        mst_cost = get_mst(remaining_list)
        min_edge = np.min(distance_matrix[node, remaining_list])

        hybrid_bound = step_cost + min_edge + mst_cost
        if hybrid_bound >= min_total:
            continue
```

```python
        # Dynamic lookahead depth
        quality_ratio = (min_total - hybrid_bound) / min_total if min_total !=
    np.inf else 1.0
        lookahead = lookahead_base + int(quality_ratio * 3)

        # Phase 3: Progressive lookahead
        next_dists = distance_matrix[node, remaining_list]
        lookahead_nodes =
    np.array(remaining_list)[np.argsort(next_dists)[:lookahead]]

        for next_node in lookahead_nodes:
            path_cost = step_cost + distance_matrix[node, next_node]
            mst_remaining = get_mst(list(remaining - {next_node}))
            lower_bound = path_cost + mst_remaining

            if lower_bound >= min_total:
                continue

            # Fast path completion
            temp_cost = path_cost
            temp_remaining = remaining - {next_node}
            last_node = next_node

            while temp_remaining:
                closest = min(temp_remaining, key=lambda x:
    distance_matrix[last_node, x])
                temp_cost += distance_matrix[last_node, closest]
                if temp_cost >= min_total:
                    break
                last_node = closest
                temp_remaining.remove(closest)

            if temp_cost < min_total:
                total_cost = temp_cost + distance_matrix[last_node,
    destination_node]
                if total_cost < min_total:
                    min_total = total_cost
                    best_node = node

    return best_node
```

## TSP Constructive Heuristic (HSEvo)

**Code**

```python
import numpy as np

def select_next_node(current_node, destination_node, unvisited_nodes,
    distance_matrix):
    if len(unvisited_nodes) == 1:
        return next(iter(unvisited_nodes))

    # Core weights with optimized 3% progress scaling
    progress = 1 - len(unvisited_nodes)/len(distance_matrix)
    w_proximity = 0.55 * (1 - 0.03*progress)
    w_cluster = 0.30 * (1 + 0.03*progress)
    w_destination = 0.15
```

```python
    max_dist = np.max(distance_matrix) or 1
    scores = {}

    # Precompute metrics
    proximity = {node: distance_matrix[current_node][node]/max_dist for node in
    unvisited_nodes}
    destination = {node: distance_matrix[node][destination_node]/max_dist for node
    in unvisited_nodes}

    # Robust cluster metric
    cluster = {}
    for node in unvisited_nodes:
        others = unvisited_nodes - {node}
        if others:
            dists = [distance_matrix[node][u]/max_dist for u in others]
            cluster[node] = np.mean(dists) - 0.5 * np.std(dists) if len(dists) > 1
    else np.mean(dists)
        else:
            cluster[node] = 0

    # Base scoring
    for node in unvisited_nodes:
        scores[node] = (w_proximity * proximity[node]
                        - w_cluster * cluster[node]
                        - w_destination * destination[node])

    # Focused lookahead for top 3 candidates
    if len(unvisited_nodes) > 3:
        top_candidates = sorted(unvisited_nodes, key=lambda x: scores[x])[:3]
        for node in top_candidates:
            remaining = unvisited_nodes - {node}
            next_node = min(remaining, key=lambda x: distance_matrix[node][x])
            lookahead = 0.7 * distance_matrix[node][next_node]/max_dist + \
                        0.3 * distance_matrix[next_node][destination_node]/max_dist
            scores[node] = 0.9 * scores[node] + 0.1 * (-lookahead)

    # Strict tie-breaking (score -- proximity -- index)
    min_score = min(scores.values())
    candidates = [n for n in scores if scores[n] == min_score]

    if len(candidates) > 1:
        min_prox = min(proximity[n] for n in candidates)
        candidates = [n for n in candidates if proximity[n] == min_prox]
        if len(candidates) > 1:
            return min(candidates)

    return candidates[0]
```

## TSP Constructive Heuristic (MCTS-AHD)

**Code**

```python
def select_next_node(current_node, destination_node, unvisited_nodes,
    distance_matrix):
    min_score = float('inf')
    next_node = -1
    progress_factor = len(unvisited_nodes) / (len(distance_matrix) - 1)
```

```
    for node in unvisited_nodes:
        immediate_dist = distance_matrix[current_node][node]
        remaining_nodes = unvisited_nodes - {node}
        if remaining_nodes:
            future_distances = [distance_matrix[node][x] for x in remaining_nodes]
            sorted_distances = sorted(future_distances)
            median_future_dist = sorted_distances[len(sorted_distances) // 2]
            avg_future_dist = sum(future_distances) / len(remaining_nodes)
            dest_dist = distance_matrix[node][destination_node]
            dynamic_weight = 1.0 + progress_factor
            penalty = (1 / max(1, median_future_dist)) * (1 + progress_factor)
            proximity_penalty = sum(1 / max(1, distance_matrix[node][u]) for u in
remaining_nodes)
            score = (immediate_dist * dynamic_weight) - (avg_future_dist /
dynamic_weight) + penalty + (0.05 * proximity_penalty) - (0.5 * dest_dist * (1
- progress_factor))
        else:
            score = immediate_dist + distance_matrix[node][destination_node]
        if score < min_score:
            min_score = score
            next_node = node
    return next_node
```

## G.9.2. OTHER HEURISTICS

### Online BPP Constructive Heuristic

**Description** Smooth adaptive hybrid scoring with dynamic tiered thresholds and variance-stabilized exact-fit detection that balances bin utilization awareness with decisive packing actions.

**Code**

```python
import numpy as np

def priority(item, bins_remain_cap):
    # Stabilized exact-fit detection with adaptive tolerance
    cap_variance = np.var(bins_remain_cap)
    epsilon = max(0.015 * item * (1 - np.tanh(cap_variance * 5)), 0.003)
    exact_fits = np.abs(bins_remain_cap - item) < epsilon

    # Dynamic threshold tiers based on utilization spectrum
    norm_caps = bins_remain_cap / max(bins_remain_cap)
    threshold_scale = 0.5 * (1 + np.sin(np.pi * (norm_caps - 0.5)))

    # Three-tier adaptive sigmoid scoring
    layer_thresh = [item * (1.04 + 0.06 * threshold_scale),
                    item * (1.16 + 0.08 * threshold_scale),
                    item * (1.35 + 0.10 * threshold_scale)]

    layer1 = 1.0 / (1.0 + np.exp(-7.0*(bins_remain_cap - layer_thresh[0])))
    layer2 = 1.0 / (1.0 + np.exp(-4.5*(bins_remain_cap - layer_thresh[1])))
    layer3 = 1.0 / (1.0 + np.exp(-2.8*(bins_remain_cap - layer_thresh[2])))

    # Dynamic weight balancing
    weight_base = np.array([0.55, 0.35, 0.10])
    weight_mod = 1.0 - 0.2 * cap_variance
    weights = np.clip(weight_base * weight_mod, 0.05, 0.65)
```

```python
    # Combined scoring
    combined_score = (layer1 * weights[0] +
                      layer2 * weights[1] +
                      layer3 * weights[2])

    # Final priority with exact-fit dominance
    priority_scores = np.where(exact_fits, 3.3, combined_score)
    priority_scores[bins_remain_cap < item] = -np.inf

    return priority_scores
```

## 0–1 KP Constructive Heuristic

**Description** This algorithm introduces a hierarchical scoring system that first clusters items by weight-value ratios, then applies dynamic capacity-based selection within each cluster, optimizing for both immediate value and future packing potential.

**Code**

```python
import numpy as np

def select_next_item(remaining_capacity, weights, values):
    if remaining_capacity <= 0:
        return -1

    # Calculate value densities
    densities = values / weights
    mean_density = np.mean(densities)

    # Cluster items into three groups based on density
    high_density = np.where(densities > mean_density * 1.5)[0]
    medium_density = np.where((densities <= mean_density * 1.5) &
                              (densities > mean_density * 0.75))[0]
    low_density = np.where(densities <= mean_density * 0.75)[0]

    clusters = [high_density, medium_density, low_density]

    best_item = -1
    best_score = -np.inf

    for cluster in clusters:
        if len(cluster) == 0:
            continue

        # Calculate dynamic weight threshold for this cluster
        cluster_weights = weights[cluster]
        weight_threshold = min(remaining_capacity,
                               np.percentile(cluster_weights, 75) *
                               (0.5 + 0.5 * (remaining_capacity / np.max(weights))))

        # Evaluate items in this cluster
        for idx in cluster:
            if weights[idx] > remaining_capacity:
                continue

            # Base score combining value and density
```

```
            score = values[idx] + 2 * densities[idx]

            # Capacity utilization bonus
            if weights[idx] <= weight_threshold:
                score += (weight_threshold - weights[idx]) * 0.5

            # Future packing potential assessment
            remaining = remaining_capacity - weights[idx]
            if remaining > 0:
                future_score = 0
                for j in np.argsort(-densities):
                    if j != idx and weights[j] <= remaining:
                        future_score += values[j]
                        remaining -= weights[j]
                        if remaining <= 0:
                            break
                score += future_score * 0.3

            if score > best_score:
                best_score = score
                best_item = idx

    return best_item
```

## TSP GLS Heuristic

**Description** An edge penalty heuristic based on normalized distance-weighted node centrality measures that prioritizes edges connecting high-traffic nodes with long distances.

**Code**

```
import numpy as np

def heuristics(distance_matrix):
    # Calculate node centrality as inverse of average distance to other nodes
    centrality = 1 / (np.mean(distance_matrix, axis=1) + 1e-10)

    # Create edge penalty matrix by multiplying distance with centrality product
    penalty_matrix = distance_matrix * (centrality[:, None] * centrality[None, :])

    # Apply logarithmic scaling to dampen extreme values
    penalty_matrix = np.log1p(penalty_matrix)

    # Normalize to [0,1] range with min-max scaling
    min_val = np.min(penalty_matrix)
    max_val = np.max(penalty_matrix)
    normalized_matrix = (penalty_matrix - min_val) / (max_val - min_val + 1e-10)

    return normalized_matrix
```

## TSP ACO Heuristic

**Description** This algorithm refines the parent approach by implementing decoupled temperature modulation and power-law scaling with entropy-constrained adaptive weighting, using geometric averaging of local statistics and simplified parameter adaptation to maintain stability while improving exploration.

**Code**

```python
import numpy as np
from scipy.special import softmax

def heuristics(distance_matrix):
    epsilon = 1e-10
    n = distance_matrix.shape[0]

    # Simplified power-law transformation with size adaptation
    base_exp = 1.1 + 0.1 * np.log(n) / np.log(50)
    smoothed_dist = distance_matrix + epsilon * np.median(distance_matrix)
    heuristic_matrix = 1 / (smoothed_dist ** base_exp)
    np.fill_diagonal(heuristic_matrix, 0)

    # Stable neighborhood parameters
    min_neighbors = max(3, int(np.log2(n)))
    max_neighbors = min(12, n//4)
    base_neighbors = min(max_neighbors, max(min_neighbors, int(n**0.35)))

    # Geometric statistics collector
    def geo_mean(x):
        return np.exp(np.mean(np.log(x + epsilon)))

    for i in range(n):
        distances = distance_matrix[i]

        # Robust neighborhood selection
        k = base_neighbors
        nearest = np.argpartition(distances, k)[:k+1]
        nearest = nearest[nearest != i][:k]

        if len(nearest) > 0:
            # Geometric combination of local statistics
            local_scale = geo_mean(distances[nearest])

            # Simplified Gaussian-Entropy weighting
            sigma = 0.5 * local_scale
            gauss_weights = np.exp(-(distances**2)/(2*(sigma**2 + epsilon)))

            # Decoupled entropy regularization
            scaled_dist = distances / (local_scale + epsilon)
            entropy_probs = softmax(-scaled_dist * np.log(k + 1))
            entropy_probs[i] = 0

            # Geometric combination preserves stability
            combined_weights = np.sqrt(gauss_weights * entropy_probs)
            heuristic_matrix[i] *= combined_weights

    # Conservative normalization
    row_max = np.max(heuristic_matrix, axis=1, keepdims=True)
    return heuristic_matrix / (row_max + epsilon)
```

# H. Examples of Evolution

# I. Prompts

This section presents the prompt templates used by **RefineEvo** to coordinate a multi-agent closed loop. All prompts are written with explicit placeholders (e.g., `# population statistics`, `# operators descriptions`, `# parent_infos`) that are instantiated at runtime using the current-generation search state, including population-level statistics, operator-library descriptions, parent algorithms, and retrieved experiences. To support reliable downstream automation, we constrain the Planner and Reflector to produce **structured outputs** (JSON enclosed in a code block) with fields such as decisions, rationales, and confidence scores. This design reduces ambiguity from free-form text and ensures that key information produced in one generation can be consistently consumed in subsequent generations.

## I.1. Prompts of Planner

The Planner decides how to allocate search effort in each generation by balancing exploration and exploitation and selecting which evolutionary operators to apply. The *Operator Selection* prompt takes population statistics and recent trends as input and outputs a structured plan including a strategy type, brief metric-based justification, a list of selected operators, and a confidence score, which can be directly used to schedule operator calls under a fixed evaluation budget.

---

**Prompt for Operator Selection**

You are an expert meta-optimization strategist specializing in evolutionary algorithm orchestration. Your role is to dynamically balance exploration and exploitation in algorithm evolution by selecting optimal operators based on population dynamics.

TASK: Operator Selection for Next Generation
**Based on the current population statistics and historical trends, determine the optimal evolutionary strategy and select operators that best fit the situation.**

POPULATION STATISTICS
Generation #t Population Statistics:
# population size # best objective # mean objective # std deviation
# diversity score # convergence rate

AVAILABLE OPERATORS
Current Description: # op description
Performance Status: # performance status
History Length: # history length generations

OUTPUT FORMAT
**Provide your response as a JSON array in a ```json code block:**

```
{
  "strategy": "exploration" or "exploitation",
  "reasoning": "Concise explanation (2-3 sentences) citing specific metrics.",
  "selected_operators": ["op1", "op2", "op3", "op4", "op5"],
  "confidence": 0.0-1.0
}
```

---

In addition, the Planner includes two prompts for online operator improvement. First, the *Improvement Checker* analyzes an operator's recent performance signals and outputs whether refinement is needed and its urgency. Second, when refinement is triggered, the *Operator Refining* prompt rewrites the operator description by making either incremental or radical adjustments based on historical effectiveness, current search dynamics, and characteristics of strong individuals.

**Prompt for Improvement Checker**

You are an expert algorithm evolution analyst specializing in operator performance evaluation and adaptive improvement decisions. Your role is to analyze operator historical performance and determine whether operators need improvement to enhance evolutionary algorithm efficiency.

TASK: Evaluate Whether Operator Needs Improvement
**Analyze the operator's historical performance and current population context to determine if this operator should be improved.**

OPERATOR INFORMATION
Operator ID: # op id
Current Description: # op description

PERFORMANCE HISTORY
Performance Status: # performance status
History Length: # history length generations

POPULATION CONTEXT
# population statistics

OUTPUT FORMAT
**Provide your response as a JSON object in a ` ``json code block:**

```
{
  "needs_improvement": true or false,
  "reasoning": "Clear explanation citing specific patterns and metrics.",
  "confidence": 0.0-1.0,
  "improvement_urgency": "low" or "medium" or "high"
}
```

**Important: Base your decision on concrete evidence from the performance history, not speculation.**

**Prompt for Operator Refining**

You are an expert in evolutionary algorithm design and meta-optimization. Your role is to analyze operator performance and intelligently adapt operator descriptions to improve algorithm evolution effectiveness.

TASK: Improve Operator # op id
**This operator has shown # performance status over the last # history length generations.**

CURRENT OPERATOR DESCRIPTION
# operators descriptions

PERFORMANCE HISTORY
Performance Status: # performance status
History Length: # history length generations

POPULATION CONTEXT
# population statistics

TOP PERFORMING ALGORITHMS
**Algorithm 1 (Objective:** # obj**)**
**Description:** # Its Description
**Algorithm 2 (Objective:** # obj**)**
**Description:** # Its Description
**...**
OUTPUT FORMAT
**Provide your response as a JSON object in a ```json code block:**

```
{
  "improvement_type": "incremental" or "radical",
  "reasoning":"Detailed explanation of why this type of improvement is needed",
  "new_description": "The improved operator description",
  "confidence": 0.0-1.0
}
```

### I.2. Prompts of Evolver

The Evolver is responsible for generating concrete algorithmic individuals. It includes prompts for both initialization and evolution. The *Initialization* prompt produces a feasible initial solution procedure, optionally guided by a provided reference implementation and external domain knowledge to improve validity and starting quality. The *Evolution* prompt generates offspring by applying the selected operator to parent algorithms, while also incorporating retrieved successful experiences and failed experiences.

**Prompt for Initialization**

You are a world-class expert in optimization algorithms and heuristic design with deep expertise in operations research, combinatorial optimization, and metaheuristic methods.

TASK: Generate Initial Algorithm
**You are tasked with designing a novel heuristic algorithm from scratch. This is the initial design phase where you will create the foundation for the evolutionary algorithm development process.**

REFERENCE IMPLEMENTATION
**Below is a baseline implementation to illustrate the expected code structure and format:**
# seed function

ADDITIONAL DOMAIN KNOWLEDGE
# external knowledge

**Prompt for Evolution**

You are a world-class expert in optimization algorithms and heuristic design with deep expertise in operations research, combinatorial optimization, and metaheuristic methods.

TASK: Evolve Algorithm Using Existing Solutions and Experiences
**You are tasked is to** # operators description

PARENT ALGORITHMS
**Below are the existing algorithms that serve as the foundation for your new design:**
**Parent Algorithm 1**
**Description:** # Its Description
**Code:** # Its Python Code Implementation
**Parent Algorithm 2**
**Description:** # Its Description
**Code:** # Its Python Code Implementation

DESIGN EXPERIENCES FROM PREVIOUS ATTEMPTS
SUCCESSFUL EXPERIENCES
**Learn from these positive examples – they represent effective design patterns:**
# successful experiences
FAILED EXPERIENCES
**Avoid these mistakes and pitfalls – they led to poor performance:**
# failed experiences

## I.3. Prompts of Reflector

The Reflector converts each "operator → parent(s) → offspring → performance change" trajectory into reusable, structured experience entries. The input is provided as a unified `TRAJECTORY DATA` JSON object, ensuring reflections remain grounded in their search context. We use two complementary reflection prompts: (i) *Reflect Successful Experience* and (ii) *Reflect Failed Experience* Both prompts output JSON arrays with summaries, suggestions, and applicable conditions, enabling consistent storage and retrieval in the experience pool for subsequent generations.

**Prompt for Reflect Successful Experience**

You are an expert algorithmic researcher specializing in reflective learning and knowledge distillation for optimization heuristics. Your expertise lies in analyzing algorithm design attempts, extracting actionable insights, and formulating reusable design patterns.

ANALYSIS TASK: Successful Trajectory
**The following algorithm evolution attempt successfully improved performance. Your task is to analyze what worked well and distill actionable insights.**

TRAJECTORY DATA
**Below are the existing algorithms that serve as the foundation for your new design:**
```json

{
    "operator_goal": # operator_desc,
    "parents": # parent_infos,
    "offspring": {"objective": #,"algorithm": #},
    "objective_type": # objective_type,
    "best_parent_objective": # baseline,
```

```
    "improvement": # improvement,
    "problem_name": # problem_name
}
```

OUTPUT REQUIREMENTS
**Provide your analysis as a JSON array with 2-3 insights. Each insight must have:**
**- summary: One-sentence key lesson (max 100 characters)**
**- recommendations: List of 2-4 specific, actionable suggestions**
**- applicable when: Clear description of when to use this insight**

**Output ONLY the JSON array in a code block. No other text.**

## Prompt for Reflect Failed Experience

You are an expert algorithmic researcher specializing in reflective learning and knowledge distillation for optimization heuristics. Your expertise lies in analyzing algorithm design attempts, extracting actionable insights, and formulating reusable design patterns.

ANALYSIS TASK: Failed Trajectory
**The following algorithm evolution attempt failed to improve performance. Your task is to analyze what went wrong and extract cautionary lessons.**

TRAJECTORY DATA
**Below are the existing algorithms that serve as the foundation for your new design:**
```json

{
    "operator_goal": # operator_desc,
    "parents": # parent_infos,
    "offspring": {"objective": #,"algorithm": #},
    "objective_type": # objective_type,
    "best_parent_objective": # baseline,
    "improvement": # improvement,
    "problem_name": # problem_name
}
```

OUTPUT REQUIREMENTS
**Provide your analysis as a JSON array with 2-3 insights. Each insight must have:**
**- summary: One-sentence key lesson about what to avoid (max 100 characters)**
**- recommendations: List of 2-4 specific warnings or alternative approaches**
**- applicable when: Clear description of when this pitfall is likely**

**Output ONLY the JSON array in a code block. No other text.**

