# OpenReview forum: "RefineEvo: Planning-Guided Heuristic Evolution with Bidirectional Experience"
_ICML.cc/2026/Conference — ICML 2026 regular_

### Official Review · Reviewer_ZaKu · 2026-02-27

**Soundness:** 3
**Presentation:** 3
**Significance:** 3
**Originality:** 2
**Overall Recommendation:** 5
**Confidence:** 4

**Summary:**

This paper propose a planning-guided, experience-driven system to enhance Automatic Heuristic Design. This paper is novel, as previous a lot of works have explore how to enhance the heuristic based current heuristic, it propose a new direction that why not reflect the operator itself. The experiment show that this framework improve the performance and efficiency.

**Compliance With Llm Reviewing Policy:**

Affirmed.

**Final Justification:**

The author solve all my concern, and I raise my score.

**Key Questions For Authors:**

1. Have the authors analyzed the behavior of the reflector module? In my view, GPT-4o-mini may not be sufficiently strong, as it sometimes produces superficial or uninformative summaries. It would be helpful if the authors could quantify how often positive experiences in BEP lead to improved results in subsequent iterations, as well as how often negative experiences still lead to improvements.

2. In addition, given the limited reasoning capability of GPT-4o-mini, the authors should consider evaluating stronger reasoning models. A more capable model may produce more meaningful experimental summaries and potentially lead to improved performance.

3. With the development of LLM, do the authors consider the direction that give up population or other evolution component, and test the results on your own methods? I think that testing on more strong model, it may can make sense.

4. This paper lack the experiment on other LLM to test the robustness of this framework. And this paper should test on CVRPLib.

5. Can this paper generalize to more difficult problem, like CVRPTW? And why not do construct method for CVRP? It is any problem for evolving constructive code for CVRP?

**Limitations:**

yes

**Strengths And Weaknesses:**

1. This paper's result is impressive, it show that RefineEvo surpass existing LLM-based model, such as EOH, ReEvo, MCTS-AHD. But it only tests on the GPT-4o-mini, I think test on other LLM model is necessary as it can test the robustness of this framework.
2. The motivation of this paper also good, reflecting the operator and refine the operator is a good direction. But this framework still in the evolution framework which is useful in the LLM that is not sufficiently strong.
3. The presentation of this paper is good.

---

> ### Author Rebuttal · Authors · 2026-03-31
>
> We thank the reviewer for the positive evaluation. We are glad the reviewer finds our results impressive, the motivation sound, and the presentation clear. We address each concern below.
>
> **Q1: The reviewer asks about the behavior of the Reflector module and whether GPT-4o-mini is sufficiently strong.**
>
> We would like to mention that our default backbone is DeepSeek-V3 (Section 4.1), while GPT-4o-mini results are reported separately in Appendix F.1 (Table 13), where RefineEvo still achieves the best performance (e.g., 8.712 vs. MCTS-AHD's 8.773 on TSP100).
>
> Regarding Reflector behavior, we now report BEP utilization statistics. Evolutionary search is inherently difficult; in our TSP experiments the final BEP (after pruning) contains ~210 entries: 22 positive and 188 negative. We tracked retrieval outcomes:
>
> - Positive experiences: 21.5% of retrievals led to offspring improving over the parent
> - Negative experiences: 12.0% of retrievals led to offspring improving over the parent
>
> The higher success rate of positive experiences confirms that they provide more actionable guidance. As shown in Table 5, removing negative experiences (13.94%) hurts more than removing positive ones (12.92%), suggesting that negative experiences play an important complementary role in preventing repeated failure modes.
>
> **Q2: The reviewer suggests evaluating stronger reasoning models.**
>
> We appreciate this suggestion and have evaluated with GPT-4o:
>
> | Method | TSP100 | TSP200 | TSP500 | BPP 5k | BPP 10k | BPP 100k |
> |---|---|---|---|---|---|---|
> | EoH | 8.724 | 12.194 | 19.129 | 0.48% | 0.39% | 0.25% |
> | ReEvo | 8.569 | 11.929 | 18.584 | 0.71% | 0.25% | 0.04% |
> | RefineEvo | **8.534** | **11.919** | **18.503** | **0.43%** | **0.22%** | **0.01%** |
>
> Combined with DeepSeek-V3 (main paper) and GPT-4o-mini (Appendix F.1), RefineEvo has been validated on three backbones of varying capability, confirming the gains stem from framework design rather than a specific LLM.
>
> **Q3: The reviewer asks whether evolutionary components could be simplified as LLMs become stronger.**
>
> We analyzed the improvement from initialization to final evolved results on the TSP100 training set across three backbones:
>
> | Backbone | Init mean | Evolved mean | Mean improv. | Init best | Evolved best | Best improv. |
> |---|---|---|---|---|---|---|
> | DeepSeek-V3 | 10.803 | 8.572 | 20.65% | 9.317 | 8.536 | 8.38% |
> | GPT-4o-mini | 10.426 | 8.675 | 16.79% | 9.370 | 8.643 | 7.77% |
> | GPT-4o | 10.609 | 8.595 | 18.99% | 9.283 | 8.518 | 8.24% |
>
> The evolutionary process improves the best individual by 7.77%-8.38% across all backbones. This substantial gap persists regardless of backbone capability, indicating the program search space remains too large for single-shot generation. These results suggest that evolutionary search remains important even with stronger backbones, though identifying the minimal necessary structure is an interesting direction for future work.
>
> **Q4: The reviewer requests experiments on other LLMs and CVRPLib.**
>
> As noted above, RefineEvo has been validated on three LLM backbones. Regarding CVRPLib, we have conducted generalization experiments:
>
> CVRPLib Constructive (gap):
> | Method | gap |
> |---|---|
> | EoH | 40.08% |
> | ReEvo | 35.72% |
> | MCTS-AHD | 40.71% |
> | PartEvo | 34.70% |
> | OpenEvolve | 32.62% |
> | RefineEvo | **29.34%** |
>
> CVRPLib ACO (gap):
> | Method | gap |
> |---|---|
> | EoH | 33.33% |
> | ReEvo | 31.91% |
> | HSEvo | 26.62% |
> | MCTS-AHD | 34.36% |
> | OpenEvolve | 21.13% |
> | RefineEvo | **16.91%** |
>
> RefineEvo achieves the best generalization under both paradigms. The advantage is more pronounced under ACO (16.91% vs. OpenEvolve's 21.13%) than under constructive heuristics (29.34% vs. 32.62%).
>
> **Q5: The reviewer asks about generalization to CVRPTW and constructive CVRP.**
>
> We have conducted experiments on both tasks:
>
> CVRP Constructive (Obj):
> | Method | N=50 | N=100 | N=200 | N=500 |
> |---|---|---|---|---|
> | PartEvo | 13.393 | 23.226 | 40.307 | 88.096 |
> | OpenEvolve | 12.837 | 22.311 | 39.493 | 88.086 |
> | RefineEvo | **12.385** | **21.630** | **38.041** | **84.601** |
>
> VRPTW Constructive (Obj):
> | Method | N=20 | N=50 | N=100 | N=200 |
> |---|---|---|---|---|
> | PartEvo | 10.828 | 20.471 | 35.157 | 58.489 |
> | OpenEvolve | 10.054 | 19.339 | 33.226 | 56.183 |
> | RefineEvo | **9.862** | **19.018** | **32.628** | **54.746** |
>
> RefineEvo achieves the best results across all scales on both tasks. In principle, any VRP variant with existing heuristic components can be integrated with LLM-based AHD for potential enhancement. We note that constructive heuristics remain a relatively weak paradigm (29.34% gap on CVRPLib), but under ACO RefineEvo reduces this to 16.91%, demonstrating that evolving components within stronger frameworks yields substantially better results.
>
> We thank the reviewer for the constructive suggestions which have strengthened our evaluation. We would be grateful if the reviewer could reconsider the score in light of these additions.

---

> > ### Author Rebuttal · Reviewer_ZaKu · 2026-04-01
> >
> > The rebuttal is good, I think it solve most of my concern. I have a few additional questions, and I believe clarifying them would raise my score.
> >
> > 1. The paper reports that 21.5% of positive assessments and 12% of negative assessments contribute to improving the framework. Could the authors clarify how this ratio should be interpreted? Does this suggest that only a subset of assessments is informative? Additionally, what is the effect of the remaining assessments, and do introduce noise?
> >
> > 2. There are more latest version language model, like cluade opus 4.6. Can the author test on it to see the performance, and also look at without GA framework, will this model still provide good results.

---

> > > ### Author Response · Authors · 2026-04-04
> > >
> > > We thank the reviewer for the thoughtful follow-up questions and for acknowledging that our rebuttal resolved most concerns. We address each question below.
> > >
> > > **Q1: How to interpret the 21.5%/12.0% retrieval success rates and whether remaining assessments introduce noise.**
> > >
> > > In LLM-based AHD, improvement is inherently rare. To quantify this, we measured the overall improvement rate (i.e., the fraction of evolutionary attempts where offspring outperform the parent) across methods:
> > > | Method | Backbone | Improvement Rate |
> > > |--------|----------|-----------------|
> > > | EoH | DeepSeek-V3 | 7.8% |
> > > | RefineEvo | DeepSeek-V3 | 13.8% |
> > > | RefineEvo | Claude Opus 4.6 | 19.6% |
> > >
> > > EoH (without experience mechanism) achieves only 7.8%; RefineEvo increases this to 13.8% under the same backbone. Against such a low-success background, the 21.5%/12.0% retrieval-conditioned rates confirm that *retrieved experiences are informative rather than noise*. The gap between 21.5% and 12.0% is likely related to pool asymmetry and their different functional roles in search: positive experiences are rarer (~22 vs. ~188 negative entries) and typically more actionable, while negative experiences serve more as constraints against repeated failures.
> > >
> > > *Non-improving retrievals are not equivalent to noise.* (1) Negative experiences constrain search away from known failure modes even without immediate improvement. (2) Our credit-assignment mechanism (Appendix E.4) prunes entries with utility below zero, removing genuinely misleading experiences. As shown in Table 5, removing the pool entirely (w/o Experience) degrades TSPLIB from 11.54% to 15.91%, confirming the cumulative guidance value.
> > >
> > > **Q2: Testing on Claude Opus 4.6 and whether the GA framework remains necessary.**
> > >
> > > We have conducted experiments with Claude Opus 4.6.
> > >
> > > **(1) Evolution improvement.** Initialization vs. evolved results on the TSP100 training set:
> > >
> > > | Backbone | Init best | Evolved best | Best improv. | Init mean | Evolved mean | Mean improv. |
> > > |----------|-----------|--------------|-------------|-----------|--------------|-------------|
> > > | GPT-4o-mini | 9.370 | 8.643 | 7.77% | 10.426 | 8.675 | 16.79% |
> > > | DeepSeek-V3 | 9.317 | 8.536 | 8.38% | 10.803 | 8.572 | 20.65% |
> > > | Claude Opus 4.6 | 9.106 | 8.024 | 11.88% | 10.137 | 8.036 | 20.72% |
> > >
> > > Claude Opus 4.6's initialization advantage is modest (best: 9.106 vs. 9.317 for DeepSeek-V3 and 9.370 for GPT-4o-mini), but after evolution it achieves the largest best-individual improvement (11.88%), substantially exceeding other backbones. This validates the insight in Q2 that *"A more capable model may produce more meaningful experimental summaries and potentially lead to improved performance"*, and demonstrates that stronger models benefit even more from evolutionary search.
> > >
> > > **(2) Cross-scale generalization.**
> > >
> > > | Backbone | TSP100 (Obj↓) | TSP200 (Obj↓) | TSP500 (Obj↓) | TSPLIB (Gap%↓) |
> > > |----------|--------|--------|--------|--------|
> > > | GPT-4o-mini (evolved) | 8.712 | 12.178 | 19.101 | 16.573% |
> > > | DeepSeek-V3 (evolved) | 8.541 | 11.868 | 18.476 | 11.545% |
> > > | Claude Opus 4.6 (evolved) | 8.018 | 11.202 | 17.478 | 5.452% |
> > > | Claude Opus 4.6 (init best, no evolution) | 9.321 | 13.013 | 19.828 | 17.751% |
> > >
> > > Even GPT-4o-mini after evolution outperforms Claude Opus 4.6's best initialization across all scales, confirming that evolutionary search remains highly beneficial. Performance improves monotonically with model capability, with Claude Opus 4.6 achieving a TSPLIB gap of only 5.452%.
> > >
> > > **(3) Cost considerations.** Claude Opus 4.6 (USD 25/M output tokens) costs ~22.7× more than DeepSeek-V3 (USD 1.10/M output tokens) in output-token pricing. Since evolutionary search involves repeated LLM calls over many generations, this pricing gap is practically important. Yet DeepSeek-V3 with evolution (TSPLIB: 11.545%) substantially outperforms Claude Opus 4.6 without evolution (TSPLIB: 17.751%), making cost-efficient backbones with evolutionary search a practical and effective choice.
> > >
> > > These results demonstrate that stronger backbones and evolutionary search are complementary rather than substitutional.
> > >
> > > We have added these results to the revised manuscript and would be grateful if the reviewer could reconsider the score in light of these additional clarifications and experiments.

---

### Official Review · Reviewer_dEYY · 2026-02-28

**Soundness:** 3
**Presentation:** 3
**Significance:** 3
**Originality:** 2
**Overall Recommendation:** 4
**Confidence:** 5

**Summary:**

This paper introduces RefineEvo, an autonomous multi-agent framework that transforms Automatic Heuristic Design (AHD) into a planning-guided, experience-driven evolution system. To overcome the rigidity of fixed evolutionary operators and the underutilization of historical search data , RefineEvo employs a Planner to dynamically schedule and refine operators based on real-time population statistics and a Bidirectional Experience Pool (BEP) to distills both successful strategies and failure pitfalls from parent-to-offspring trajectories into situation-conditioned guidance. Experimental results across diverse combinatorial optimization benchmarks demonstrate that RefineEvo consistently outperforms baselines.

**Compliance With Llm Reviewing Policy:**

Affirmed.

**Final Justification:**

After a thorough review of the original submission and the authors’ comprehensive rebuttal, I have updated my evaluation from Weak Reject to Weak Accept. The authors have successfully addressed my primary concerns regarding baseline competitiveness, the necessity of core components through expanded ablation studies, and the qualitative differentiation of their method.

**Key Questions For Authors:**

See weakness.
- Are positive and negative samples treated with equal priority within the BEP? Specifically, how does the framework handle the cases where a theoretically sound algorithmic innovation performs poorly due to a minor implementation error in the LLM-generated code?

**Limitations:**

yes

**Strengths And Weaknesses:**

**Strengths**:
1. The paper is well-motivated, identifying two critical bottlenecks in current LLM-based AHD: the inefficiency of static, "use-all" operator scheduling and the lack of situational grounding in experience accumulation.
2. The framework demonstrates a superior performance across diverse classic CO problems, achieving best results while improving token efficiency.
3. The paper is clearly written with a logical structure.

**Weakness**:
1. Baseline: While the authors compare RefineEvo against many LLM-based methods, the handcrafted heuristics baselines are somewhat simplistic and not competitiveness. Additionally, the absence of a comparison with recent frameworks like AlphaEvolve/OpenEvolve leaves a gap in the competitive landscape.
2. Performance: Although RefineEvo claims SOTA status, the absolute performance delta in some tasks is minimal. In Table 1, the objective values for KP across different methods show differences often smaller than 0.1%. Reporting results in terms of the optimality gap might provide a clearer picture of the actual significance of these improvements.
3. Ablation: The ablation results raise questions regarding the individual necessity of the core components. For example, in Table 4&5, the "w/o Experience" variant (8.687 on TSP100) and the "w/o refinement" variant (8.702) both appear to perform competitively against all the LLM-based baselines. These results potentially overshadowing the claimed impact of the Planner or BEP.
4. Token Efficiency: While the authors heavily emphasize token efficiency in the abstract, experiments, and conclusion , this objective is almost entirely absent from the Introduction and Methodology. This lack of alignment between the initial problem framing and the final claims may undermine the logical consistency of the narrative.
5. Lack of Content Analysis: The experimental section focuses heavily on quantitative metrics. However, there is a missed opportunity to provide a deeper component-level analysis of the generated heuristics. A detailed comparison of the specific code structures or mathematical terms produced by RefineEvo versus baselines would help clarify why the generated heuristics are qualitatively superior.

---

> ### Author Rebuttal · Authors · 2026-03-31
>
> We thank the reviewer for the detailed evaluation. We are encouraged by the recognition of our motivation in identifying key bottlenecks in LLM-based AHD, the strong performance across diverse problems, and the clarity of writing. We address each concern below.
>
> **W1: The reviewer suggests strengthening baselines by including AlphaEvolve/OpenEvolve.**
>
> As stated in Section 4.1, handcrafted baselines follow the standard protocol of prior AHD works. We have added OpenEvolve and PartEvo under identical settings:
>
> TSP Constructive (Obj):
> |Method|N=100|N=200|N=500|TSPLIB|
> |---|---|---|---|---|
> |PartEvo|8.701|12.137|18.854|13.44|
> |OpenEvolve|8.639|12.074|18.792|12.92|
> |RefineEvo|**8.541**|**11.868**|**18.476**|**11.54**|
>
> BPP (excess-bin ratio):
> |Method|5k_C100|10k_C100|100k_C100|
> |---|---|---|---|
> |PartEvo|0.34%|0.22%|0.02%|
> |OpenEvolve|0.28%|**0.16%**|**0.01%**|
> |RefineEvo|**0.25%**|0.19%|**0.01%**|
>
> CVRP Constructive (Obj):
> |Method|N=50|N=100|N=200|N=500|
> |---|---|---|---|---|
> |PartEvo|13.393|23.226|40.307|88.096|
> |OpenEvolve|12.837|22.311|39.493|88.086|
> |RefineEvo|**12.385**|**21.630**|**38.041**|**84.601**|
>
> VRPTW Constructive (Obj):
> |Method|N=20|N=50|N=100|N=200|
> |---|---|---|---|---|
> |PartEvo|10.828|20.471|35.157|58.489|
> |OpenEvolve|10.054|19.339|33.226|56.183|
> |RefineEvo|**9.862**|**19.018**|**32.628**|**54.746**|
>
> RefineEvo achieves the strongest overall performance across these added benchmarks. We have also evaluated on CVRPLib under both constructive (29.34% vs. OpenEvolve 32.62%) and ACO (16.91% vs. 21.13%) settings; due to space limits here, full CVRPLib results are deferred to our response to Reviewer ZaKu and the revised appendix.
>
> **W2: The reviewer suggests reporting the optimality gap for KP.**
>
> Reporting absolute values follows prior convention. We agree that the gap is clearer and have converted all KP results in the revised paper.
>
> **W3: The reviewer notes that ablation variants still appear competitive on TSP100.**
>
> TSP100 is an in-distribution benchmark where most methods achieve similar performance. The critical impact becomes evident on harder settings:
>
> | Variant | TSP100 | TSPLIB (OOD) | BPP 5k_C100 |
> |---|---|---|---|
> | RefineEvo | **8.541** | **11.54%** | **0.25%** |
> | w/o Experience | 8.687 (+1.7%) | 15.91% (+37.9%) | 0.38% (+52.0%) |
> | w/o Refinement | 8.702 (+1.9%) | 15.63% (+35.4%) | 0.30% (+20.0%) |
> | Best baseline (MCTS-AHD) | 8.732 | 14.53% | 0.31% |
>
> Both w/o variants fall below MCTS-AHD on TSPLIB and BPP, showing these components are critical to RefineEvo's advantage on complex, out-of-distribution settings.
>
> **W4: Token efficiency is emphasized in abstract/experiments but insufficiently discussed in Introduction and Methodology.**
>
> The central bottleneck in LLM-based AHD is not only solution quality but also search efficiency under repeated LLM calls, making token efficiency a natural part of the problem framing. As reported in Figure 4(b), RefineEvo achieves the best trade-off (~511K tokens, comparable to HSEvo ~497K, but with better quality). We have strengthened this in the Introduction and added a breakdown in the appendix.
>
> **W5: The reviewer suggests qualitative analysis of generated heuristic code structures.**
>
> As provided in Appendix F.5.1, generated heuristics for all methods are included. We have added qualitative comparison: (1) RefineEvo produces multi-stage pipelines (dynamic filtering, regret-based re-inclusion, adaptive-depth rollout, composite scoring), while EoH/MCTS-AHD uses flat single-stage evaluation. (2) RefineEvo's candidate control uses dynamic thresholding with regret-based re-inclusion, and its rollout depth adapts to search progress (2 to 4 steps). In contrast, HSEvo applies fixed 1-step lookahead on top-3 candidates, while MCTS-AHD relies on statistical approximation without explicit rollout. (3) RefineEvo uniquely integrates destination-awareness into candidate filtering via a destination ratio that shapes the threshold, enabling earlier pruning. These structural differences may explain RefineEvo's stronger scalability on larger instances.
>
> **Q1: Whether positive and negative experiences have equal priority, and how implementation errors are handled.**
>
> As described in Section 3.3, both types are stored and retrieved independently via semantic relevance, without fixed priority. The utility-based pruning (Appendix E.4) adjusts pool composition: experiences failing to improve downstream outcomes are pruned regardless of polarity. If a sound idea fails due to a coding error, the Reflector may record a negative experience; however, the "applicable_when" field captures specific context, so reattempting under different conditions reduces incorrect retrieval. The BEP treats negatives not as universal prohibitions, but as condition-bound records whose utility is continuously re-evaluated.
>
> We would be grateful if the reviewer could reconsider the assessment in light of these clarifications and additions.

---

> > ### Author Rebuttal · Reviewer_dEYY · 2026-04-02
> >
> > Thanks for your response.

---

> > > ### Author Response · Authors · 2026-04-03
> > >
> > > We sincerely thank you for the careful review and valuable suggestions. Your insightful feedback has significantly strengthened our paper. We are encouraged by your positive reassessment.

---

### Official Review · Reviewer_4duq · 2026-03-04

**Soundness:** 3
**Presentation:** 3
**Significance:** 3
**Originality:** 3
**Overall Recommendation:** 5
**Confidence:** 4

**Summary:**

The authors propose RefineEvo, an LLM based evolutionary AHD framework that dynamically selects operators as well as making adjustments to them, and maintains a context-aware experience library which keeps track of both good and bad results.

**Compliance With Llm Reviewing Policy:**

Affirmed.

**Final Justification:**

The rebuttal addressed my previous concerns, I will keep my score, Accept. This is my final recommendation.

**Key Questions For Authors:**

1. What are the detailed hyperparameters and initialization prompts for each method?
2. Is it possible to evaluate the effectiveness of semantics based query?
3. How is the requirement of number of tokens and time in each iteration compared to other methods?

**Limitations:**

YES

**Strengths And Weaknesses:**

Strength: The paper is technically sound with detailed experiment results and fair comparison with other methods. The presentation is clear and easy to follow, and the difference between the proposed method and prior work is clear. For significance and originality, the proposed framework addresses two important drawbacks of prior LLM based AHD methods, introducing new mechanics naming dynamic evolution prompt and situation-aware experience.

Weakness: My major concern is the scalability and practical impact of LLM-based AHD methods. Take the TSP as an example: traditional algorithms and solvers have already handled instances with thousands or even millions of nodes.

---

> ### Author Rebuttal · Authors · 2026-03-31
>
> We thank the reviewer for the positive evaluation. We are encouraged by the recognition of the technical soundness of our work, the fairness of our experimental comparisons, the clarity of presentation, and the significance of our framework. We address each concern below.
>
> **W1: Scalability and practical impact of LLM-based AHD methods.**
>
> We appreciate this concern and would like to clarify two important points. First, the LLM is used only during the offline heuristic design phase. Once evolved, the heuristic is a lightweight Python function applicable to instances of any scale without LLM involvement. Second, as demonstrated in our ACO and GLS experiments (Section 4.2), RefineEvo evolves key components within established metaheuristic frameworks, directly enhancing existing solvers already capable of handling large-scale instances.
>
> More broadly, the goal of AHD is not to replace carefully engineered large-scale solvers, but to reduce the design cost of key heuristic components and to automatically discover high-quality strategies that can be transferred into existing solver pipelines. We believe extending this paradigm to improve components within state-of-the-art solvers is a promising and practical direction.
>
> **Q1: Detailed hyperparameters and initialization prompts for each method.**
>
> As documented in Appendix D (Table 9) of our submission, all RefineEvo hyperparameters are provided, including population size (N=10), generations (T=30), temperature (1.0), top-k retrieval (k=3), and embedding dimension (1024). The complete operator prompt library is provided in Appendix E.2 (Table 10), showing initial descriptions for all five meta-operators and examples of how they evolve through refinement. For baselines, we preserved each method's original search configuration and used their official open-source implementations, controlling the comparison primarily through a unified evaluation budget of 1000 evaluations, following common practice in AHD comparisons (EoH, ReEvo, MCTS-AHD). We have strengthened the cross-referencing between the main text and the appendix in the revised paper.
>
> **Q2: Effectiveness of semantic-based query retrieval.**
>
> We appreciate this suggestion and have conducted an ablation comparing three retrieval strategies:
>
> | Strategy | TSP100 | TSP200 | TSP500 | BPP 5k_C100 | BPP 10k_C100 | BPP 100k_C100 |
> |---|---|---|---|---|---|---|
> | Random | 8.742 | 12.247 | 19.528 | 0.43% | 0.28% | 0.03% |
> | TF-IDF | 8.657 | 12.107 | 19.081 | 0.36% | 0.20% | **0.01%** |
> | Semantic (ours) | **8.541** | **11.868** | **18.476** | **0.25%** | **0.19%** | **0.01%** |
>
> Random retrieval performs worst due to noise from irrelevant experiences. TF-IDF improves by capturing surface lexical overlap but cannot distinguish experiences with similar keywords yet opposite recommendations. Semantic retrieval outperforms both by capturing deeper contextual similarity, particularly the structured applicability conditions. We have added this ablation to the appendix of the revised paper.
>
> **Q3: Token and time cost per iteration compared to other methods.**
>
> Figure 4(b) of our submission shows the overall performance-token trade-off. We further provide detailed token statistics below:
>
> | Method | Total completion tokens | Tokens per evaluation |
> |---|---|---|
> | HSEvo | ~497K | ~497 |
> | PartEvo | ~507K | ~507 |
> | RefineEvo | ~511K | ~511 |
> | MCTS-AHD | ~585K | ~585 |
> | EoH | ~595K | ~595 |
> | ReEvo | ~751K | ~751 |
>
> RefineEvo's per-evaluation cost (~511 tokens) is among the lowest, comparable to HSEvo and PartEvo but with substantially better solution quality. For wall-clock time, direct cross-method comparison is less reliable because heuristic evaluation dominates runtime and different methods use different parallelization settings. We therefore report tokens as the primary reproducible cost metric, and provide RefineEvo's own per-iteration time breakdown:
>
> | Component | Time(s) | Proportion |
> |---|---|---|
> | Operator selection (Planner) | 2.41 | 10.35% |
> | Heuristic generation (LLM) | 11.49 | 49.33% |
> | Semantic retrieval (encoding + search) | 0.36 | 1.55% |
> | Reflection (LLM) | 9.03 | 38.77% |
>
> Semantic retrieval accounts for only 0.36s (1.53%), and the experience-related cost is dominated by the Reflector's LLM call. We have added both breakdowns to the appendix of the revised paper.
>
>
> We thank the reviewer for the supportive evaluation and constructive questions. We have followed the suggestions to add retrieval ablation, token breakdown, and strengthened appendix references to improve the paper.

---

> > ### Author Rebuttal · Reviewer_4duq · 2026-04-01
> >
> > Thank you for the response.

---

> > > ### Author Response · Authors · 2026-04-03
> > >
> > > We sincerely thank you for the supportive evaluation and thoughtful questions. Your insightful suggestions have significantly strengthened our paper.

---

### Official Review · Reviewer_H2r8 · 2026-03-06

**Soundness:** 3
**Presentation:** 3
**Significance:** 3
**Originality:** 2
**Overall Recommendation:** 4
**Confidence:** 5

**Summary:**

This paper introduces RefineEvo, a improved LLM-based automatic heuristic design framework for combinatorial optimization. Its core contributions lie in two aspects: (1) dynamically selecting and refining evolutionary operators (prompts) based on the current search state to guide the evolution process, and (2) proposing a Bidirectional Experience Pool (BEP) that integrates both successful strategies and failure patterns to inform future evolution directions. Experiments on several classic combinatorial optimization benchmarks (e.g., TSP, KP, BPP) demonstrate that RefineEvo consistently outperforms strong baselines in terms of solution quality and token efficiency.

**Compliance With Llm Reviewing Policy:**

Affirmed.

**Final Justification:**

The proposed dynamic operator adjustment is a timely contribution to the field of automated heuristic design. I previously questioned the competitiveness of these operators against expert-designed baselines, as the original draft focused on early-stage LLM-driven methods.

The rebuttal clarified this by comparing the performance against PartEvo. Given that PartEvo represents a strong baseline with carefully engineered strategies, the results successfully demonstrated the robustness of the authors' approach. Coupled with the statistical and overhead analyses, my concerns have been fully addressed. I recommend acceptance, **with the strict proviso that the new experimental data provided during the rebuttal is fully integrated into the final version**.

**Key Questions For Authors:**

The main text tables do not seem to comprehensively report the performance of FunSearch and HSEvo on the TSP benchmarks. Could the authors clarify the reason for omitting these comparisons in the main tables and elaborate on how RefineEvo compares to them on these specific instances?

**Limitations:**

See Weaknesses above

**Strengths And Weaknesses:**

**Strengths**

1. The study of dynamic operators represents a relatively novel direction in the field of LLM+EC, making this work highly timely and relevant.

2. The paper is well structured, clearly written, and easy to follow.

3. The ablation studies on the primary contributing components are comprehensive and effectively demonstrate the performance improvements they bring to the framework.

**Weaknesses**

1. Contribution Distinguishability: Although the paper proposes Dynamic Operator Selection and Refinement, as well as the Reflector and BEP, the Reflector component shares similarities with the reflective evolution mechanism in ReEvo[1], and the core idea of bidirectional experience storage in BEP has also been preliminarily explored in PartEvo [2]. LLM-LNS [3] has also studied the dynamic operator approach. The paper requires stronger empirical evidence to prove that dynamic operators can substantially surpass manually crafted operators to demonstrate sufficient impact.

2. Insufficient Validation of Dynamic Operator Refinement: Given that pioneering works like FunSearch and EoH have relatively preliminary operator designs, outperforming them is expected. I highly recommend the authors include a direct comparison with PartEvo (which is also integrated into the LLM4AD platform). PartEvo features carefully designed prompt-centric and EC-inspired operators. If RefineEvo can outperform PartEvo, it would straightforwardly and convincingly demonstrate that dynamically adjusted operators are superior to human-designed ones.

3. Incomplete Experimental Statistical Validation: For evolutionary algorithms, randomness can easily lead to significant performance fluctuations. It is absolutely essential to provide statistical analysis (e.g., variance or standard deviation) over multiple independent runs to rigorously verify the stability of the proposed algorithm.

4. Lack of Overhead Analysis: The paper does not mention the storage size of the BEP or the time overhead associated with semantic retrieval. I suggest adding relevant statistical data in the appendix and analyzing the specific impact of the BEP on the overall runtime efficiency of the framework.

[1]Ye, Haoran, et al. "Reevo: Large language models as hyper-heuristics with reflective evolution." Advances in neural information processing systems 37 (2024): 43571-43608.

[2]Hu, Qinglong, and Qingfu Zhang. "Partition to evolve: Niching-enhanced evolution with llms for automated algorithm discovery." The Thirty-ninth Annual Conference on Neural Information Processing Systems. 2025.

[3] Ye, Huigen, et al. "Large language model-driven large neighborhood search for large-scale milp problems." Forty-second International Conference on Machine Learning. 2025.

---

> ### Author Rebuttal · Authors · 2026-03-31
>
> We thank the reviewer for the thorough evaluation and for recognizing the novelty of dynamic operators, the clarity of our presentation, and the comprehensiveness of our ablation studies. We address each concern below.
>
> **W1: Could the authors clarify the distinctions between RefineEvo and related mechanisms in ReEvo, PartEvo, and LLM-LNS, and why these differences matter empirically?**
>
> As discussed in Sections 2.1-2.2 and 3.2-3.3, we clarify the distinctions as follows.
>
> **vs. ReEvo/PartEvo (Experience).** ReEvo merges short-term insights into a single long-term text each generation, risking dilution and injecting all reflections regardless of relevance. PartEvo's RE reflects on each parent without performance comparison; its SE summarizes the overall archive landscape to identify promising regions, but does not trace which specific modifications led to improvement. Neither extracts negative experiences nor binds insights to applicability conditions. RefineEvo's BEP addresses these gaps: (1) trajectory-aware parent-to-offspring extraction, (2) bidirectional storage of successes and pitfalls, and (3) situation-conditioned retrieval. As reported in Table 5, removing trajectory (15.32%), negatives (13.94%), or situational grounding (14.33%) all degrade significantly vs. full RefineEvo (11.54%).
>
> **vs. LLM-LNS (Dynamic Operators).** LLM-LNS uses rule-based stagnation detection with a fixed threshold K=3 regardless of search state. RefineEvo replaces this with LLM-based state-aware planning. As shown in Table 4, Fixed-Interval Refinement achieves 14.05% vs. RefineEvo's 11.54% on TSPLIB.
>
> These distinctions are further validated by the PartEvo/OpenEvolve comparisons in W2.
>
> **W2: The reviewer suggests comparing with PartEvo to validate that dynamically adjusted operators surpass carefully designed fixed ones.**
>
> We appreciate this helpful suggestion. We have compared RefineEvo with PartEvo and OpenEvolve (open-source AlphaEvolve reproduction) under identical settings.
>
> TSP Constructive (Obj, mean(std)):
> |Method|N=100|N=200|N=500|TSPLIB|
> |---|---|---|---|---|
> |PartEvo|8.701(0.093)|12.137(0.116)|18.854(0.180)|13.44(0.128)|
> |OpenEvolve|8.639(0.042)|12.074(0.053)|18.792(0.067)|12.92(0.046)|
> |RefineEvo|**8.541(0.031)**|**11.868(0.034)**|**18.476(0.049)**|**11.54(0.023)**|
>
>
> BPP (excess-bin ratio(std)):
> |Method|5k_C100|10k_C100|100k_C100|
> |---|---|---|---|
> |PartEvo|0.34%(0.05)|0.22%(0.02)|0.02%(0.01)|
> |OpenEvolve|0.28%(0.03)|**0.16%(0.01)**|**0.01%(0.01)**|
> |RefineEvo|**0.25%(0.03)**|0.19%(0.02)|**0.01%(0.01)**|
>
> RefineEvo consistently outperforms PartEvo and achieves strong overall results, especially on TSP and TSPLIB, while remaining competitive on BPP.
>
> **W3: The reviewer suggests reporting standard deviations.**
>
> Following the reviewer's suggestion, we now explicitly report mean(std) for the main results averaged over 3 independent runs. As shown above, RefineEvo consistently achieves the smallest standard deviation (e.g., 0.031 on TSP100 vs. PartEvo's 0.093), indicating better stability.
>
> **W4: The reviewer requests an overhead analysis of BEP storage and retrieval time.**
>
> As described in Section 3.3, the BEP employs utility-based pruning to keep the pool concise. We have added a detailed breakdown to the appendix.
>
> After pruning, the BEP retains ~210 experience entries, occupying ~750KB for text content and ~7MB for embeddings, negligible in practice.
>
> Per-iteration time breakdown:
> |Component|Time(s)|Proportion|
> |---|---|---|
> |Operator selection (Planner)|2.41|10.35%|
> |Heuristic generation (LLM)|11.49|49.33%|
> |Semantic retrieval (encoding + search)|0.36|1.55%|
> |Reflection (LLM)|9.03|38.77%|
>
> Semantic retrieval (query embedding + cosine similarity over ~210 vectors) accounts for only 0.36s (1.55%). The experience-related cost is dominated by the Reflector's LLM call for extracting structured insights. As shown in Table 5, removing experiences degrades from 11.54% to 15.91% on TSPLIB, indicating this overhead is well compensated by the performance gain.
>
> **Q1: The reviewer asks about the reporting of FunSearch and HSEvo on TSP benchmarks.**
>
> As reported in our submission, both are included in constructive (Table 1, N=50-1000) and ACO (Figure 3) TSP results. The only gap was GLS (Table 3), because their original papers did not evaluate on GLS, and prior GLS studies (EoH, ReEvo, MCTS-AHD) also did not include them as baselines, so we followed this established convention. Following this suggestion, we have adapted and evaluated them on GLS:
>
> GLS on TSP (gap%):
> | Method | TSP50 | TSP100 | TSP200 |
> |---|---|---|---|
> | FunSearch | 0.0067 | 0.0088 | 0.3997 |
> | HSEvo | **0.0000** | 0.0011 | 0.2785 |
> | RefineEvo | **0.0000** | **0.0000** | **0.2240** |
>
> RefineEvo achieves the best or tied-best GLS performance. These results have been added to Table 3.
>
> We would be grateful if the reviewer could reconsider the assessment in light of these new results and clarifications.

---

> > ### Author Rebuttal · Reviewer_H2r8 · 2026-04-01
> >
> > Thank you for the comprehensive rebuttal. The new empirical evidence has successfully addressed all of my prior concerns. I have raised my score accordingly. Good job.

---

> > > ### Author Response · Authors · 2026-04-03
> > >
> > > We sincerely thank you for the careful and constructive review. Your insightful suggestions have significantly strengthened our paper. We are encouraged by your positive reassessment.

---

### Decision · Program_Chairs · 2026-04-30

**Decision:**

Accept (regular)

**Comment:**

The paper introduces an LLM system for automatically designing heuristics for solving combinatorial problems. All reviewers agreed that this is an important contribution to include in the ICML proceedings. The contribution is indeed timely as it reduces the cost of using traditional approaches to solving combinatorial search problems. This is an interesting trend, and it offers an alternative view to "LLMs don't plan" or "LLMs can't solve combinatorial search problems".

The technical contributions of the paper are a set of heuristics (state-aware operator selection and bidirectional experience pool) for designing heuristics to challenging combinatorial search problems such as the TSP and Knapsack. Empirical results show superior performance in two axes: search performance and token usage.